# Competition for the conserved branch point sequence influences physiological outcomes in pre-mRNA splicing

Karen Larisssa Pereira de Castro[1], Jose M Abril[1], Kuo-Chieh Liao[2], Haiping Hao[3], John Paul Donohue[4], William K Russell[3], William S Fagg[1,3]*

[1]Transplant Division, Department of Surgery, University of Texas Medical Branch, Galveston, United States; [2]RNA Genomics and Structure, Genome Institute of Singapore, Agency for Science, Technology, and Research (A*STAR) Singapore, Singapore, Singapore; [3]Department of Biochemistry and Molecular Biology, University of Texas Medical Branch, Galveston, United States; [4]Sinsheimer Labs, RNA Center for Molecular Biology, Department of Molecular, Cell and Developmental Biology, University of California, Santa Cruz, Santa Cruz, United States

## eLife Assessment

This **important** manuscript provides insights into the competition between Splicing Factor 1 (SF1) and Quaking (QKI) for binding at the ACUAA branch point sequence in a model intron, regulating exon inclusion. The study employs **convincing**, rigorous transcriptomic, proteomic, and reporter assays, with both mammalian cell culture and yeast models.

*For correspondence:
wsfagg@UTMB.EDU

Competing interest: The authors declare that no competing interests exist.

**Abstract** Recognition of the intron branch point during spliceosome assembly is a multistep process that can influence mRNA structure and levels. A branch point sequence motif UACUAAC is variably conserved in eukaryotic genomes, but in some organisms, more than one protein can recognize it. Here, we show that SF1 and Quaking (QKI) compete for a subset of intron branch sites with the sequence ACUAA in mammalian cells. SF1 activates exon inclusion through this sequence, but QKI represses the inclusion of alternatively spliced exons with this intron branch point sequence. Using mutant reporters derived from a natural intron with two branch site-like sequences, we find that when either branch point sequence is mutated, the other is utilized; however, when both are present, neither is used due to high-affinity binding and strong splicing repression by QKI. QKI occupancy at the dual branch site directly prevents SF1 binding and the subsequent recruitment of spliceosome-associated factors. Finally, ectopic expression of QKI in budding yeast (which lacks *QKI*) is lethal, at least in part due to widespread splicing repression. In conclusion, QKI can function as a splicing repressor by directly competing with SF1/BBP for a subset of branch point sequences that closely mirror its high-affinity binding site.

## Introduction

Up to 95% of human protein-coding genes can be alternatively spliced (***Wang et al., 2008***; ***Pan et al., 2008***). In contrast, only a minority of *S. cerevisiae* genes contain introns, and under normal growth conditions, most are efficiently spliced (***Spingola et al., 1999***; ***Munding et al., 2010***). Many influences converge to promote extensive alternative splicing that is observed in higher eukaryotes, including genome complexity, *cis*-acting regulatory elements, and *trans*-acting factors (***Fu and Ares,***

2014; *Ule and Blencowe, 2019*; *Marasco and Kornblihtt, 2023*). An early post-transcriptional step that is required for splicing is branch point (bp) recognition, which, along with 5′ splice site (ss), poly-pyrimidine (pY) tract, and 3′ss recognition, ultimately defines the spliceosome E-complex (*Bennett et al., 1992*; *Seraphin and Rosbash, 1989*). Sequence variability in these elements can influence the efficiency with which the E-complex and subsequent splicing complexes mature (*Wahl et al., 2009*; *Zhuang and Weiner, 1986*; *Michaud and Reed, 1991*; *Hoskins et al., 2011*; *Berglund et al., 1997*; *Zamore et al., 1992*). An interesting bp-sequence-specific observation is that different organisms maintain conservation; for example, in *Saccharomyces cerevisiae* (*S. cerevisiae*), it is nearly invariant (UACUAAC), whereas in mammals it is more degenerate (YUNAY) (*Moore et al., 1993*; *Mercer et al., 2015*; *Zeng et al., 2022*; *Gould et al., 2016*). The underlying functional factors that explain this variability in bp sequence/branch-site (bs) conservation are unknown. This suggests that bs variability might provide a way to control alternative splicing, but how this could be mediated is unclear.

RNA-binding proteins (RBPs) can influence bs recognition by SF1 (mammals) (*Berglund et al., 1997*; *Liu et al., 2001*) or MSL5/BBP (*S. cerevisiae*) (*Berglund et al., 1997*; *Jacewicz et al., 2015*). When these and other E-complex-associated proteins and snRNAs associate with the substrate, SF1/BBP is evicted and replaced by the SF3B complex, which recruits the 17S U2 snRNP, signaling maturation into the spliceosome A complex (*Kastner et al., 2019*; *Shi, 2017*; *Martínez-Lumbreras et al., 2024*). Disruption of these events by either mutations or non-physiological concentrations of RBPs can lead to splicing defects and disease (*Love et al., 2023*). For example, the balance between splicing-activating serine-arginine (SR) proteins and splicing-repressing heterogeneous nuclear ribonucleoproteins (hnRNPs) regulates a large set of pre-mRNA substrates and illustrates how competition between RBPs regulates alternative splicing (*Busch and Hertel, 2012*; *Hertel, 2008*). However, it is unclear whether RBP competition influences cell type-specific splicing or has evolutionary implications.

The metazoan RBP Signal Transduction and Activator of RNA metabolism (STAR) family members possess KH-type and QUA2 RNA-binding domains and regulate diverse forms of RNA processing, including splicing (*Artzt and Wu, 2010*). Interestingly, SF1 is the most divergent member of the STAR family, as it lacks the QUA1 dimerization domain that all other members possess (*Liu et al., 2001*; *Ryder and Massi, 2010*; *Vernet and Artzt, 1997*). The QUA1-containing members like Quaking (QKI), KHDRBS1-3, GLD1, and ASD2 exist as dimers and thus bind to a bipartite sequence motif in their target RNAs (*Galarneau and Richard, 2005*; *Galarneau and Richard, 2009*). QKI (*Teplova et al., 2013*; *Beuck et al., 2012*) and SF1 (*Liu et al., 2001*) are structurally similar (*Figure 1A*), and although SF1 lacks the QUA1 dimerization domain, they bind to a similar sequence motif. Interestingly, the SF1 binding motif (*Corioni et al., 2011*) is more degenerate than the QKI consensus motif (*Galarneau and Richard, 2005*; *Ryder and Williamson, 2004*; *Hafner et al., 2010*; *Fagg et al., 2017*; *Hayakawa-Yano et al., 2017*), which is consistent with SF1's role in the recognition of the degenerate mammalian bs.

Analysis of tissue-specific splicing patterns nearly 20 years ago revealed that the UACUAAY motif is associated with muscle-specific exon skipping when located in the intron upstream of alternatively spliced exons or exon inclusion when located in the downstream intron, but the opposite pattern was observed in the brain (*Sugnet et al., 2006*). It was unclear how the 'conserved bp sequence' UACUAAY might promote exon skipping or inclusion from the downstream intron. Still, it suggested that this element may have an additional and specialized role in tissue-specific alternative splicing. Clarity was provided for this model by the discovery that QKI directly regulates pre-mRNA splicing in C2C12 myoblasts via binding to an ACUAAY element and exhibiting the prototypical 'splicing code' positionally-based regulatory pattern (*Fagg et al., 2017*; *Hall et al., 2013*; *Barash et al., 2010*). This finding is also consistent with the observation that the nuclear isoform QKI5 is expressed at relatively high levels in muscle but at lower levels in the brain (*Fagg et al., 2017*; *Cox et al., 1999*; *Justice and Hirschi, 2010*; *Noveroske et al., 2002*; *Hardy et al., 1996*; *Lu et al., 2003*; *Ebersole et al., 1996*; *Kondo et al., 1999*). Thus, QKI5 enforces cell type-specific alternative splicing patterns by binding to a bp sequence that is also a high-affinity QKI binding site, thereby promoting exon skipping.

Studies in lung/lung cells suggested that QKI could repress exon inclusion by competing with SF1 for a bs that might also function as a QKI binding site in the *NUMB* pre-mRNA (*Zong et al., 2014*). While this raised an intriguing possibility, it was unclear if these two RBPs were in direct competition for binding to a bona fide bs, and how competition for this specialized (but conserved consensus) bs might affect global splicing patterns, modulate splicing in naturally occurring intron substrates, alter

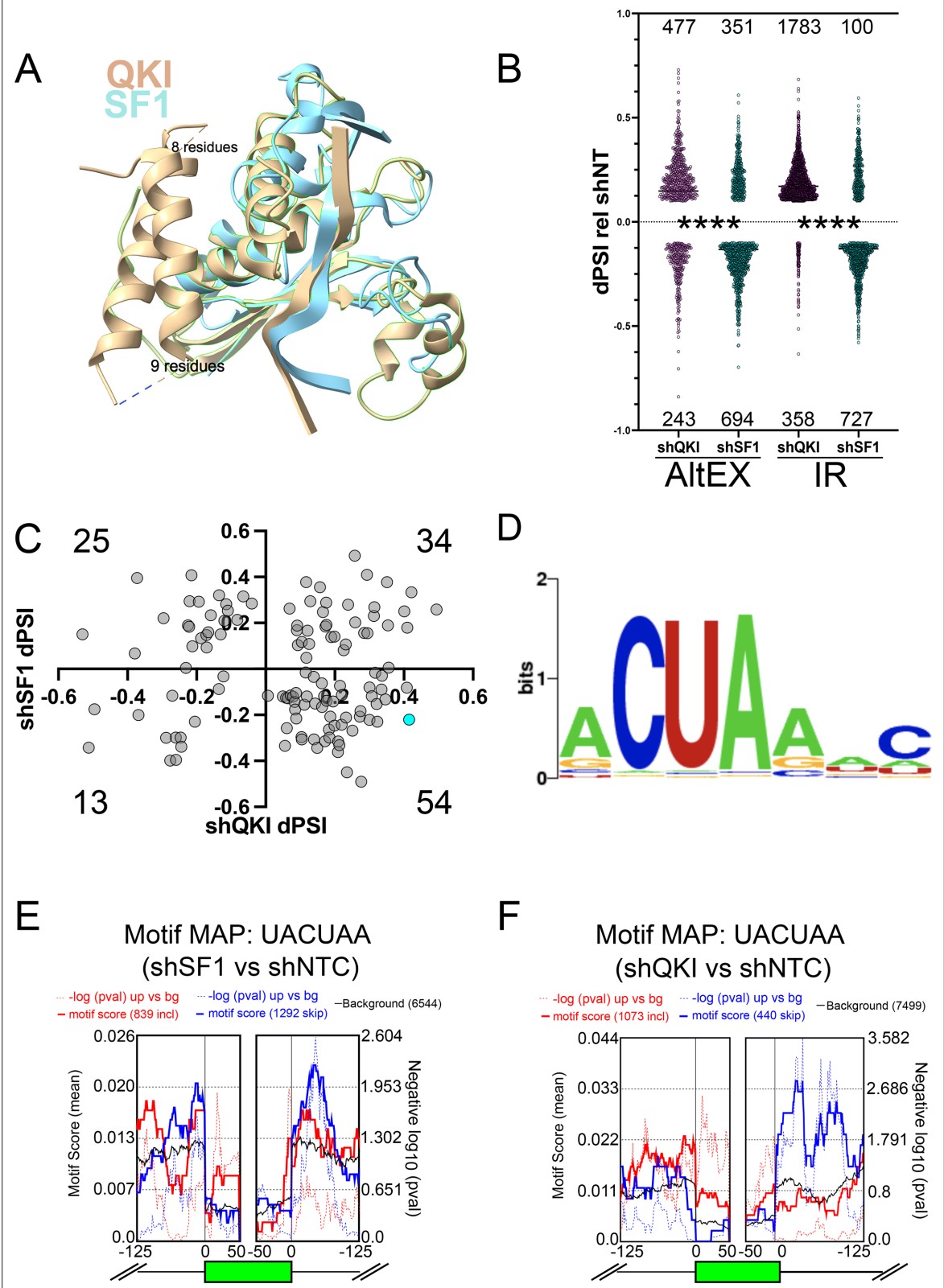

**Figure 1.** Similarity of Splicing Factor 1 (SF1) and Quaking (QKI) and RNA-seq analysis of SF1 or QKI loss-of-function in HepG2 cells. (**A**) Overlay of SF1 (cyan) and QKI (tan) KH and QUA2 protein domains: green highlighted region indicates KH and QUA2 domains of QKI; the two tan alpha helices on the left side make up the QUA1 domain in QKI (which SF1 lacks; text indicates residues in unstructured regions of QKI). (**B**) Vast-tools analysis of the ENCODE RNA-seq data from SF1 (shSF1) or QKI shRNA (shQKI) knockdown compared to control shRNA (shNT) in HepG2 cells; the y-axis shows dPSI

*Figure 1 continued on next page*

*Figure 1 continued*

for shQKI or shSF1 relative to shNT for significantly altered alternatively spliced exons (AltEX; left) or intron retention (IR; right; dPSI ≥ |10| and MVdPSI > 0), and ****p<0.0001 by Mann-Whitney U when comparing the distribution of changes in shQKI relative to shNT to shSF1 relative to shNT. (**C**) Scatter plot showing the distribution of AltEX events that changed under depletion of both SF1 (dPSI values relative to shNT on the y-axis) and QKI (dPSI values relative to shNT on the x-axis); the number in each quadrant indicates how many AltEX events were observed; *RAI14* exon 11 is shown in cyan. (**D**) Simple Enrichment Analysis (SEA) of the intron region spanning 60 nt to 20 nt upstream of the 3'ss in QKI and SF1 regulated AltEX events shown in C (*p*<0.05). (**E**) rMAPS motif map for UACUAA generated for AltEX events changing during shSF1 compared to shNT by rMATS; motif scores (solid line) or -log10 p-value (dotted line) is shown in red for exons whose inclusion increases and in blue for exons whose inclusion decreases. (**F**) rMAPS motif map as described in **E**. but for shQKI treatment compared to shNT.

spliceosome component recruitment, or influence evolutionary outcomes in splicing. Furthermore, SF1 knockdown in HeLa cells indicates it is required for viability and the splicing of several introns (***Tanackovic and Krämer, 2005***), but the extent to which its loss affects global splicing patterns is unknown. Here, we test these open questions and find that QKI5 can repress a set of alternatively spliced exons whose bs mirror the high-affinity QKI binding site. We use RAI14 exon 11 as a model and show that ACUAAC elements in intron 10 are true bs to which QKI or SF1 can bind to promote exon skipping or inclusion, respectively. Interestingly, QKI binding to RAI14 intron 10 bs leads to the recruitment of paraspeckle-associated proteins, while SF1 binding causes subsequent enrichment of the SF3a/b and U2 17 S complexes. These discoveries expand the scope and clarify the bp competition model. Moreover, the addition of QKI5 into *S. cerevisiae* is lethal and concomitant with widespread splicing repression. Together, these findings suggest that the presence of *QKI* and degenerate bs may have coevolved to expand the repertoire of cell type-specific alternative splicing in plants and multicellular eukaryotes.

## Results

### QKI and SF1 co-regulate a set of alternatively spliced exons through a distinct sequence motif

Based on analysis of their individual functions (***Berglund et al., 1997***; ***Liu et al., 2001***; ***Corioni et al., 2011***; ***Hall et al., 2013***; ***de Bruin et al., 2016***; ***van der Veer et al., 2013***; ***Darbelli et al., 2016***; ***Selenko et al., 2003***; ***Peled-Zehavi et al., 2001***) and common structural features (***Figure 1A***), we hypothesized that QKI could repress the inclusion, while SF1 could activate the inclusion of a special subset of alternatively spliced exons that have ACUAA for their bs. To test this, we analyzed RNA sequencing (RNA-seq) datasets during QKI or SF1 knockdown (HepG2 cells treated with control non-targeting shRNA or shRNAs targeting *QKI* (shQKI) or *SF1* (shSF1) ***Van Nostrand et al., 2020a***) with VAST-tools to measure changes in alternative pre-mRNA splicing (***Irimia et al., 2014***; ***Tapial et al., 2017***). Consistent with potentially opposing functions in splicing, more exon inclusion and intron retention (IR) events are observed during QKI knockdown, but more exon skipping and fewer IR events are observed during SF1 knockdown; the distribution of these changes are significantly different when compared to one another (****p<0.0001 by Mann-Whitney U; ***Figure 1B***, and ***Supplementary file 1***; significance cutoff: change in percent spliced in (dPSI) > |10| and minimum value dPSI at 95% confidence interval (MVdPSI95) >0). To measure the distribution of exons regulated by both QKI and SF1 (co-regulated exons), we plotted dPSI values of cassette exons under each condition relative to control for those that change under both knockdown conditions (dPSI > |10| in either shQKI or shSF1 relative to control, and MVdPSI95 >0 in both shQKI relative to control and SF1 relative to control). Interestingly, we observed that 54 of the 126 (43%) co-regulated alternatively spliced exons fell into the category in which inclusion increases upon QKI knockdown and decreases upon SF1 knockdown (***Figure 1C***). Thus, the most highly represented set of these (by quadrant) is QKI-repressed and SF1-activated. We next asked if these co-regulated alternatively spliced exons are associated with any significantly enriched sequences in the upstream intron region in which the bs is found. To do so, we obtained the sequence from 63 to 20 nucleotides upstream of the 3' splice site (these should include putative bs) but exclude pY tracts and 3' splice sites of these co-regulated exons or 1000 control exons from transcripts that are expressed (base mean > 100) but whose splicing is unchanged upon either QKI or SF1 knockdown (dPSI < |1|, MVdPSI = 0) and used Simple Enrichment Analysis (SEA ***Bailey and Grant, 2021***) to test if any motifs are significantly enriched. This analysis revealed

enrichment of a single ACUAA-like motif in the co-regulated exons (*p*=0.033), but failed to identify any significantly enriched motifs in the control sequences (*Figure 1D*). Interestingly, this motif could serve as a QKI (*Galarneau and Richard, 2005*; *Ryder and Williamson, 2004*; *Hafner et al., 2010*) or SF1 (*Corioni et al., 2011*; *Van Nostrand et al., 2020a*) binding site, or as a bs (*Mercer et al., 2015*; *Zeng et al., 2022*). Finally, splicing analysis of these RNA-seq datasets using the complementary method rMATS (*Shen et al., 2014*) corroborated these findings by indicating more exon inclusion when QKI is knocked down and more skipping when SF1 is reduced (*Figure 1E and F*; *Supplementary files 2 and 3*, respectively). Motif enrichment analysis using rMAPS2 (*Park et al., 2016*; *Hwang et al., 2020*) shows significant enrichment of the UACUAA motif in introns upstream of exons that are skipped more when SF1 is reduced compared to control (*Figure 1E*), and enrichment of this motif in introns upstream of exons is more included when QKI is reduced compared to control (*Figure 1F*). In summary, the most common set of exons co-regulated by SF1 and QKI is SF1-activated and QKI-repressed, and these have enrichment of a bs-like sequence in their proximal upstream intron that also appears to be a QKI binding motif.

## RAI14 exon 11 is QKI-repressed and SF1-activated, with dual putative ACUAAC branch sites

To further investigate how competition between QKI and SF1 for the bs is mediated, we sought a prototypical alternatively spliced exon subject to this form of regulation. The criteria by which we narrowed our search were (1) a QKI-repressed and SF1-activated cassette exon with, (2) experimental evidence of a functional ACUAAY bs, and (3) experimental evidence of direct QKI binding. We previously found that QKI5 represses *Rai14* exon 11 and that reads from QKI iCLIP-seq map to the intron immediately upstream of this exon in mouse myoblasts (*Fagg et al., 2017*). Its inclusion is also repressed by QKI and activated by SF1 in HepG2 cells (*Figure 1C* (cyan dot) and *Figure 2A*; *Van Nostrand et al., 2020a*). Inspection of the upstream intron region proximal to *RAI14* exon 11 revealed intriguing features that fulfilled the above criteria: tandem ACUAAC elements 34 nt or 43 nt upstream of the 3'ss, experimental evidence indicating that either of these could be used as a bs (*Zeng et al., 2022*), and QKI eCLIP peaks from experiments using HepG2 or K562 cells (*Van Nostrand et al., 2020a*; *Van Nostrand et al., 2020b*) that overlap with these putative ACUAAC bs sequences (*Figure 2A*). We next tested if the co-regulation of *RAI14* exon 11 splicing by QKI and SF1 observed in HepG2 cells could be observed in additional cell types. We measured *RAI14* exon 11 inclusion in HEK293 *QKI* KO cells and found a significant increase in its inclusion in the KO cells compared to WT (*p*<0.0001 by Student's t-test; *Figure 2B*). To reduce SF1 levels, we used two independent siRNAs targeting it in WT HEK293 cells. The first failed to produce an appreciable knockdown, but the second, or the two combined, reduced the SF1 protein level, leading to greater *RAI14* exon 11 skipping (*Figure 2C*). Overexpression of myc:QKI5 but not a myc:QKI5 construct with reduced RNA binding activity (K120A;R124A *Teplova et al., 2013*) promotes skipping of *RAI14* exon 11 in WT HEK293 cells, and overexpressing SF1 causes more inclusion (*Figure 2—figure supplement 1*). Although the physiological level of *Rai14* exon 11 inclusion is low in mouse myoblasts (perhaps due to the high level of QKI5 in these cells), knocking down *QKI5* or *SF1* in C2C12 cells leads to a significant increase or decrease, respectively, in its inclusion compared to control (*p*<0.0001 by Student's t-test; *Figure 2D*). Therefore, *RAI14* exon 11 splicing is repressed by QKI and activated by SF1 in various cell types, possibly by binding dual ACUAAC bs elements.

## Efficient RAI14 exon 11 skipping requires tandem ACUAAC elements, either of which can be used as a bp

Next, we asked how the ACUAAC elements in *RAI14* intron 10 influence exon 11 alternative splicing. Previous investigations did not stringently discriminate between whether QKI and SF1 competed directly for binding to the bs or if dimeric QKI bound to a bipartite motif that flanked the bs and thus occluded SF1 binding to the true bp (*Zong et al., 2014*). We hypothesized that: (1) efficient exon 11 skipping/splicing repression requires two intact ACUAAC elements (or at least a single ACUAAC element and QKI 'half-site' (which would constitute the previously SELEX-defined high affinity Quaking response element (QRE)) *Galarneau and Richard, 2005*), (2) one ACUAAC element or the other must be present for any exon inclusion (either can be a bs), (3) loss of both ACUAAC elements would lead to no exon inclusion (one or the other is required to have a bs), (4) conversion

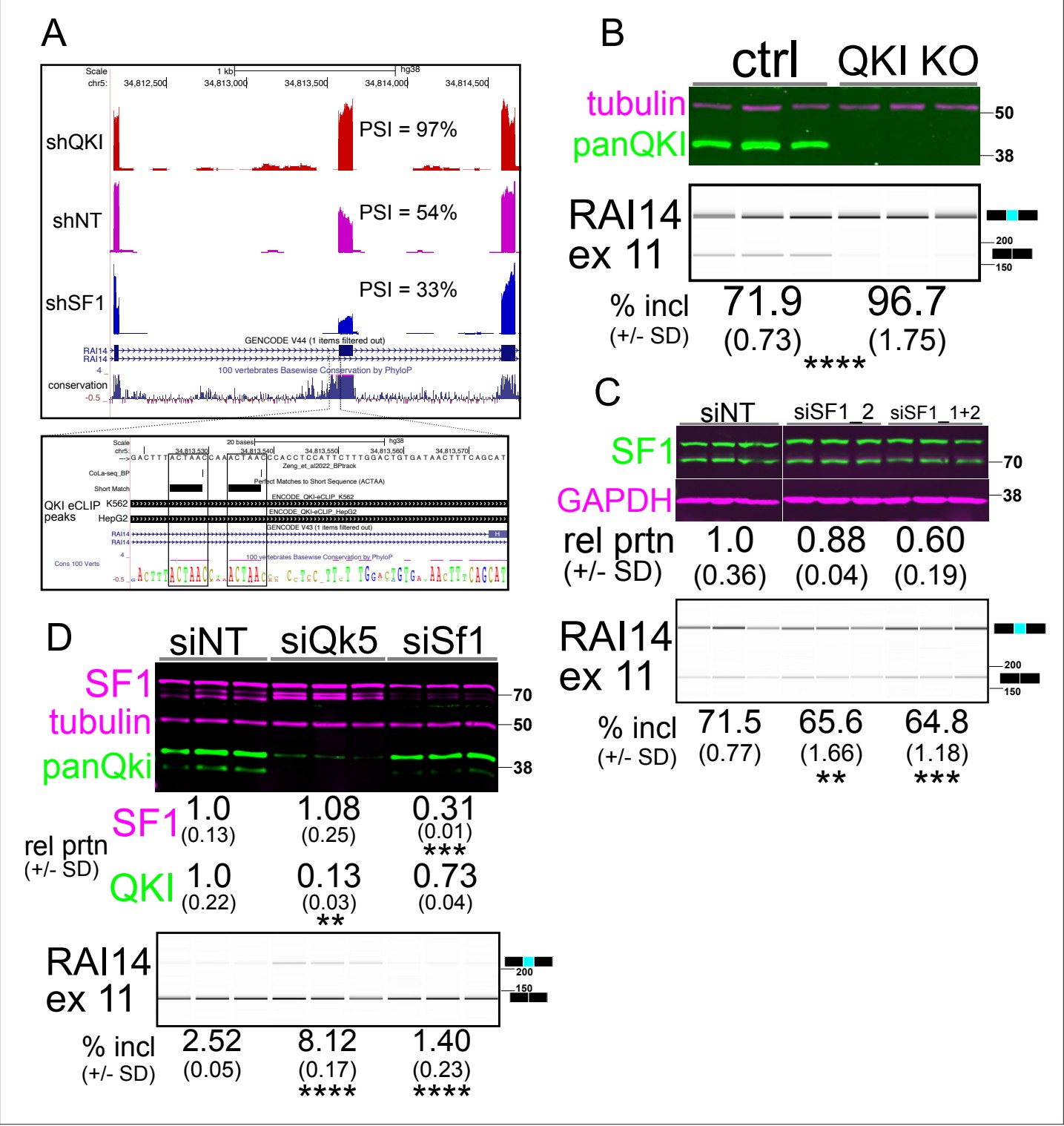

**Figure 2.** RAI14 exon 11 is repressed by QKI and activated by Splicing Factor 1 (SF1). (**A**) UCSC Genome Browser screenshot (top panel) with RNA-seq reads mapping to RAI14 for shQKI (top), shNT (middle), or shSF1 (bottom); percent spliced in (PSI) values are measured by Vast-tools; the inset shows boxed regions of (from top) two ACUAAC elements with Cola-seq branchpoints mapping to one nucleotide downstream of a branchpoint adenine, ACUAA elements by oligomatch, then Quaking (QKI) eCLIP peaks from K562 (top) or HepG2 (bottom) cells that overlap with these; conservation of 100 vertebrates is shown at the bottom. (**B**) Western blot of protein extracted from HEK293 *QKI* KO cells (top) probed with anti-tubulin (magenta) or anti-panQKI (green) antibodies (MW in kD is shown on right); RT-PCR of *RAI14* exon 11 and BioAnalyzer gel-like image (bottom) of RNA extracted from the cells above showing mean percent included ± standard deviation below (n=3 biological replicates; ****p<0.001 by Student's t-test compared to ctrl;

*Figure 2 continued on next page*

*Figure 2 continued*

size of amplicon in bp is shown on right). (**C**) Western blot (top) of proteins extracted from HEK293 cells transfected with siNT, siSF1_2, or siSF1_1+2, probed with anti-SF1 (green) or anti-Gapdh (magenta) antibodies; the protein abundance (fold change relative to the siNT control ± standard deviation) is shown below and MW in kD is shown on right; RT-PCR of *RAI14* exon 11 and BioAnalyzer gel-like image (bottom) of RNA extracted from the cells described above with mean percent included ± standard deviation below (n=3 biological replicates; **$p<0.01$ or ***$p<0.001$ by Student's t-test compared to siNT; size of amplicon in bp is shown on right). (**D**) Western blot (top) of proteins extracted from C2C12 myoblasts transfected with siNT, siQki or siSf1_1+2, probed with anti-SF1 (magenta; top), anti-tubulin (magenta; middle) or anti-panQki (green; bottom); MW in kD is shown on right. The protein abundance (fold change relative to the siNT control ± standard deviation) is shown below (n=3 biological replicates; **$p<0.01$ or ***$p<0.001$ by Student's t-test); RT-PCR of *Rai14* exon 11 and BioAnalyzer gel-like image (bottom) of RNA extracted from the C2C12 cells described above with mean percent included ± standard deviation indicated below (****$p<0.0001$ by Student's t-test; size of amplicon in bp is shown on right).

The online version of this article includes the following source data and figure supplement(s) for figure 2:

**Source data 1.** Original membranes and BioAnalyzer gel-like images corresponding to *Figure 2A* (left), *Figure 2C* (middle), or *Figure 2D* (right).

**Source data 2.** Source data containing original uncropped western blots and BioAnalyzer gel-like images/files.

**Figure supplement 1.** Western blot and RT-PCR of proteins and RNA extracted from wild-type (WT) HEK293 cells transfected with tdTomato, WT myc:Qki5, MT myc:Qki5, and Splicing Factor 1 (SF1).

**Figure supplement 1—source data 1.** Original uncropped membrane from western blot and uncropped gel-like image from BioAnalyzer shown in *Figure 2—figure supplement 1*.

**Figure supplement 1—source data 2.** Original uncropped membrane from western blot and uncropped gel-like image from BioAnalyzer shown in *Figure 2—figure supplement 1*.

of ACUAAC elements to another (non-QKI binding motif) bs would lead to more inclusion; regarding RNA stability: (5) removal of either ACUAAC element could destabilize the transcript, but (6) restoring either to a UAAC 'half-site' could rescue this defect and promote exon skipping. To test these, we generated a splicing reporter by cloning 243 nucleotides of RAI14 intron 10, RAI14 exon 11, and 95 nucleotides of RAI14 intron 11 into the beta globin splicing reporter pDUP51 (DUP-RAI14 exon 11). We first deleted either the first, the second, or both ACUAAC elements (*Figure 3A*), transfected these reporter plasmids into C2C12 cells, and then measured exon inclusion by RT-PCR or total transcript stability by RT-qPCR. Deletion of either the upstream or downstream ACUAAC element causes a ~40% increase in inclusion (*Figure 3B*), indicating that both sites are required for strong splicing repression. Interestingly, we observed the appearance of a mis-spliced product from the upstream deletion mutant compared to WT (*Figure 3B*); the latter may be due to the use of a weak/cryptic bp and 3'ss (see * on *Figure 3B*). Essentially, no inclusion of RAI14 exon 11 is observed in the absence of both ACUAAC elements, except for the mis-spliced product noted above (*Figure 3B*). Control experiments show that these amplicons are reverse transcriptase-dependent, indicating that the products detected are not due to plasmid DNA or PCR product contamination and corroborate the results described above (*Figure 3—figure supplement 1*). Moreover, loss of the upstream ACUAAC element results in markedly reduced total reporter RNA levels; we observe a similar trend in the other deletion mutants, though to a lesser extent (*Figure 3C*). Therefore, potent splicing repression of RAI14 exon 11 requires both ACUAAC elements, and one or the other is required for any exon inclusion, and thus they are bona fide bs.

We next asked whether substituting either ACUAAC element, or both, to ACUGAC (which should be a poor substrate for QKI binding but a suitable bp sequence, *Figure 3A*) would also lead to increased inclusion of RAI14 exon 11 or influence reporter RNA stability. We observe a large increase in exon inclusion and a decrease in skipping for either of the single substitution mutants and for the double substitution mutant (~20–35%; *Figure 3D*). As above, the amplification of these products is RT-dependent (*Figure 3—figure supplement 1*). Transcript abundance is significantly reduced in either single substitution mutant (****$p<0.0001$ by Student's t-test) and is slightly lower in the double substitution mutant (*Figure 3E*). Finally, converting either ACUAAC element to a UAAC 'half-site' results in comparable increases in inclusion/decreases in skipping compared to WT as described in the deletion or substitution mutants (*Figure 3F*), which are also RT-dependent (*Figure 3—figure supplement 1*). Interestingly, these 'half-site' mutants show a higher level of abundance than the WT reporter, especially in the upstream 'half-site' mutant (*Figure 3G*). In summary, removal of either ACUAAC element or conversion to ACUGAC leads to splicing activation of RAI14 exon 11, indicating a requirement for dual ACUAAC elements for potent splicing repression.

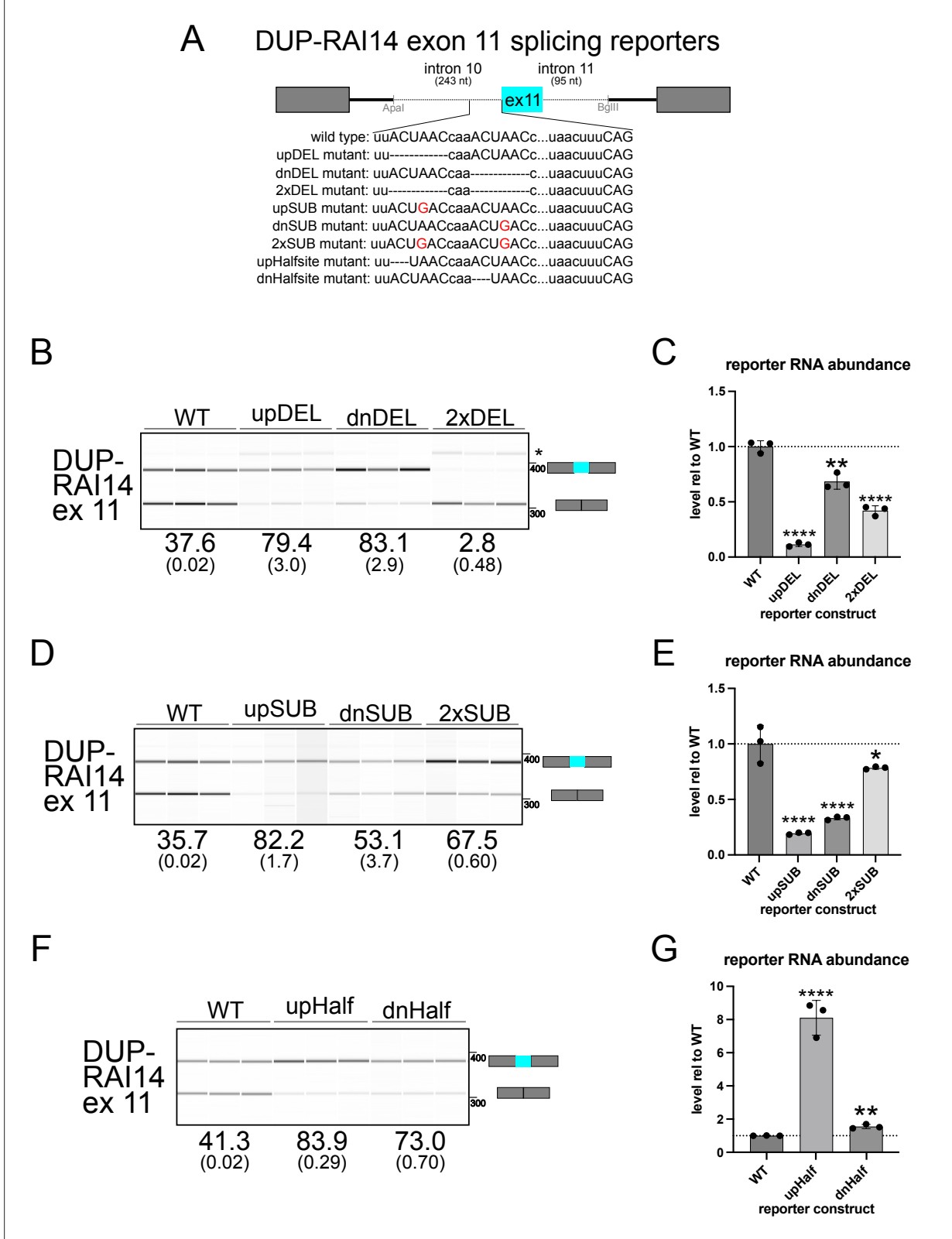

**Figure 3.** Analysis of the DUP-RAI14 exon 11 (ex 11) splicing reporter. (**A**) Schematic of the beta globin pDUP-RAI14 ex 11 splicing reporters indicating the region of intron 10, exon 11, and intron 11 included; the inset below shows the different constructs (wild-type or mutant) tested. (**B**) RT-PCR and BioAnalyzer gel-like image from RNA extracted from C2C12 cells transfected with *RAI14* ex 11 wild-type or deletion mutant reporters; mean percent included values are shown below (+/- the standard deviation; *indicates an unidentified/spurious product). (**C**) RT-qPCR measuring total reporter RNA

*Figure 3 continued on next page*

**Figure 3 continued**

level, normalized to *Eef1a1*, from RNA described in B., and shown as fold-change relative to WT (**$p<0.01$ or ****$p<0.001$ by Student's t-test). (**D**) RT-PCR and BioAnalyzer gel-like image from RNA extracted from C2C12 cells transfected with *RAI14* ex 11 wild-type or substitution mutant reporters, analyzed as described in B. (**E**) RT-qPCR measuring total reporter RNA level, normalized to *Eef1a1*, from RNA described in D., and shown as fold-change relative to WT (*$p<0.05$ or ****$p<0.001$ by Student's t-test). (**F**) RT-PCR and BioAnalyzer gel-like image from RNA extracted from C2C12 cells transfected with *RAI14* ex 11 wild-type or half-site mutant reporters, analyzed as described in B (***$p<0.001$, ****$p<0.0001$). G. RT-qPCR measuring total reporter RNA level, normalized to *Eef1a1*, from RNA described in F., and shown as fold-change relative to WT (**$p<0.01$ or ****$p<0.001$ by Student's t-test). Each experiment was conducted in biological triplicate, and for RT-PCR with BioAnalyzer measurement, each comparison compared to WT reporter showed $p<0.0001$ by Student's t-test.

The online version of this article includes the following source data and figure supplement(s) for figure 3:

**Source data 1.** Original BioAnalyzer gel-like images corresponding to *Figure 3B* (left), *Figure 3D* (middle), or *Figure 3F* (right).

**Source data 2.** Original BioAnalyzer gel-like images and data files corresponding to *Figure 3B* (left), *Figure 3D* (middle), or *Figure 3F* (right).

**Figure supplement 1.** Agarose gel showing PCR amplification in the absence (-) or presence (+) of reverse transcriptase.

**Figure supplement 1—source data 1.** Original uncropped agarose gel images from *Figure 3—figure supplement 1*.

**Figure supplement 1—source data 2.** Original uncropped agarose gel images from *Figure 3—figure supplement 1*.

## QKI binding to Rai14 intron 10 requires both ACUAAC elements and prevents spliceosome recruitment

We next hypothesized that QKI binding to Rai14 intron 10 requires both ACUAAC elements, that its binding would prevent spliceosome recruitment by blocking SF1, and that removal of either ACUAAC element would favor SF1 binding and recruitment of spliceosome components. To minimize bias, we initially performed RNA affinity chromatography (RAC) using 64 nt of intron sequence upstream of the 3'ss and including 6 nt of exonic sequence linked to a tobramycin aptamer (WT) (*Hartmuth et al., 2002*), and the same sequence but with either the upstream ACUAAC (upDEL), downstream ACUAAC (dnDEL), or both ACUAACs (2xDEL), as well as a tobramycin aptamer (APT) only RNA (*Figure 4A*). Our rationale for this was to identify all proteins bound to these substrates and correlate them with splicing outcomes: WT shows low levels of inclusion, upDEL slightly higher levels of inclusion, dnDEL high levels of inclusion, and 2xDEL lacks a bp and so is completely skipped. We added C2C12 myoblast nuclear extract (NE) to these under conditions that favor splicing (with ATP), then collected the associated proteins and identified them by liquid chromatography with tandem mass spectrometry (LC-MS/MS) (*Hartmuth et al., 2002*). Quaking is readily identified as associating with the WT RAC substrate and in the input NE but is undetectable in each of the mutant RAC substrates and the APT-only control; SF1 is not detected associating with any RAC substrate but was present in NE (*Figure 4B and C*, and *Supplementary file 4*). We used previously published LC-MS/MS datasets to generate a list of early spliceosome (E complex) and 17S U2 snRNP components (E/U2) (*Makarov et al., 2012*; *Sharma et al., 2008*; *Cvitkovic and Jurica, 2013*) in order to focus on these proteins in our RAC-LC-MS/MS datasets to test our hypotheses. Subsequently, we found that the proteomic profiles associated with the WT and 2xDEL substrates are most similar (*Figure 4B*), suggesting that QKI protein binding or loss of bs leads to similar E/U2 protein binding patterns. Indeed, in either of the single deletion mutants, we observe more enrichment of E/U2 components with the substrate RNA (24 positive and 5 negative values in upDEL and 16 positive values and seven negative values in dnDEL) while the WT and 2xDEL substrates show a more balanced distribution (18 positive and 12 negative or 18 positive and 11 negative values, respectively; *Figure 4C*). Western blot analysis using the same RAC approach also shows robust QKI association with the WT substate and undetectable or nearly undetectable SF1 protein, validating our LC-MS/MS findings (*Figure 4D*).

Previous studies indicate that excluding ATP from RNA splicing/binding assays favors a more stable association of early splicing complexes, including SF1 (*Michaud and Reed, 1991*; *Reed, 1990*; *Jamison et al., 1992*). Therefore, we performed RAC-LC-MS/MS with the same substrates in the absence of ATP (*Supplementary file 5*). We also employed a data-independent acquisition (DIA) LC-MS/MS method, utilizing NE to construct a peptide search library, thereby enhancing the specificity and sensitivity of detection (see Methods). We observe significant enrichment ($\log_2$ fold change $> |0.2|$ and $p<0.01$) of E/U2 components (including SF1) binding to the RNA in the absence of the downstream ACUAAC (dnDEL) element compared to WT RNA (*Figure 4E*). This trend is also observed but to a lesser extent in the absence of the upstream ACUAAC (upDEL) and in the absence of both

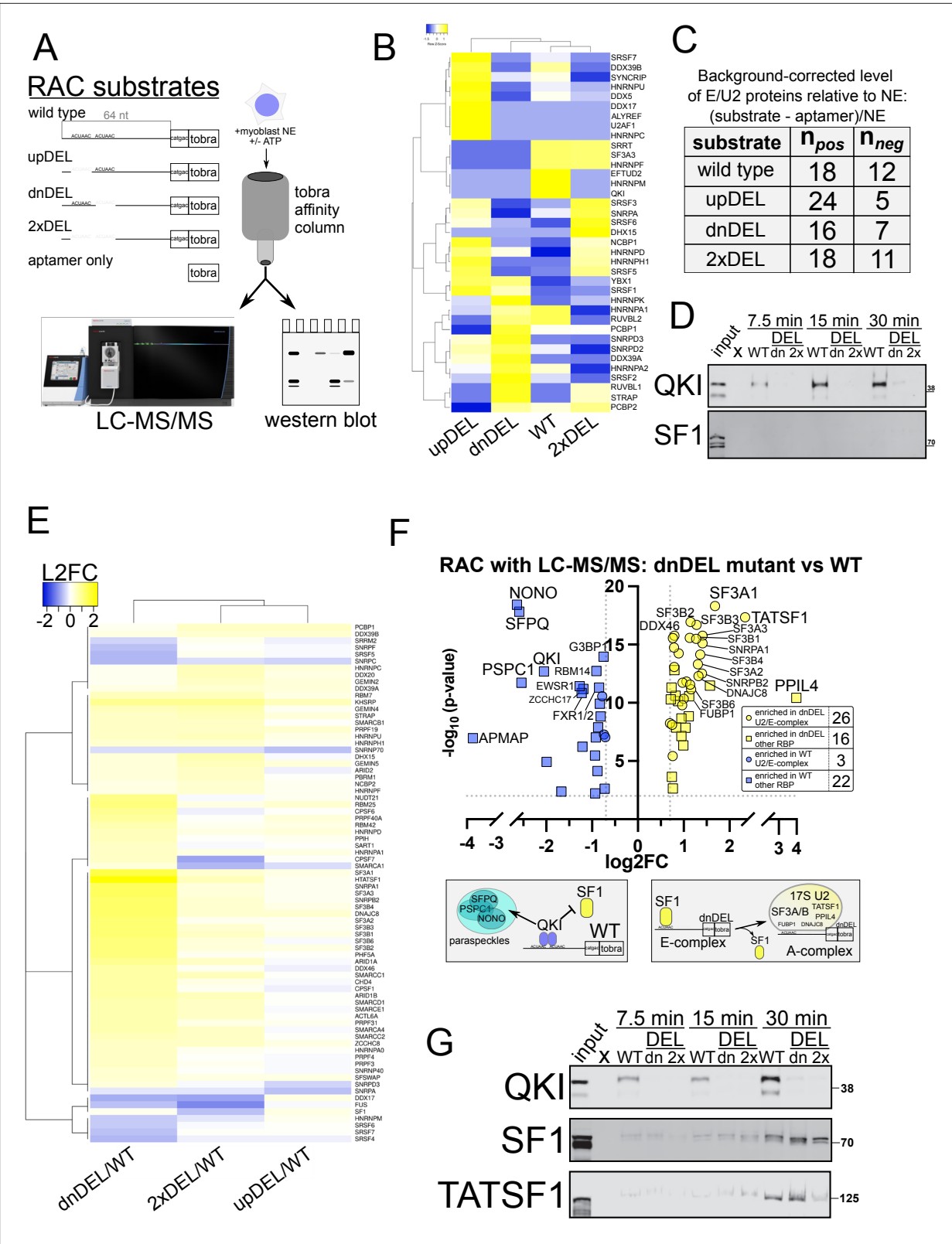

**Figure 4.** Analysis of the proteins associating with *RAI14* intron 10 RNA. RNA affinity chromatography (RAC) with liquid chromatography and tandem mass spec or western blot analysis. (**A**) Schematic representation of substrates used for RNA affinity chromatography (RAC) which included 64 nt of *RAI14* intron 10, 6 nt of exon sequence, and the tobramycin aptamer (tobra): wild-type, upDEL, dnDEL, 2xDEL, and aptamer only; C2C12 nuclear extract (NE) was incubated with these, and RAC-associated eluates were analyzed by liquid chromatography with tandem mass spectrometry (LC-MS/MS) or

*Figure 4 continued on next page*

*Figure 4 continued*

western blot. (**B**) Heatmap showing hierarchical clustering of early spliceosome and 17S U2 snRNP protein (E/U2) abundance detected in the RAC-LC-MS/MS datasets for each substate shown and in the presence of ATP; these represent background-corrected levels relative to NE (see Methods) and the scale bar shows row Z-score values. (**C**) Numbers observed for relative levels of E/U2 proteins (background corrected relative to NE) observed associating with each RAC substate with either a positive ($n_{pos}$) or negative ($n_{neg}$) value. (**D**) Western blot of NE (input) or WT, dnDEL (dn), or 2xDEL (2 x) RAC time-course (+ATP as described in B and C) for 7.5 min (left), 15 min (middle), or 30 min (right) probed with anti-panQki (top) or anti-SF1 (bottom) antibodies; MW shown in kD to the right. (**E**) Heatmap showing hierarchical clustering of early spliceosome and 17S U2 snRNP protein (E/U2) abundance detected in the RAC-LC-MS/MS datasets for each substate shown and in the absence of ATP; these represent data-independent acquisition (DIA; see methods) values normalized to NE and each mutant is shown as log2 fold change relative to the WT substate and passed cutoff of $\log_2$ fold change > |0.2| and $p<0.01$. (**F**) Volcano plot comparing LC-MS/MS $\log_2$ protein abundance ($\log_2$ fold-change (log2FC); x-axis) of E/U2 (circles) and other RNA-binding proteins (RBPs) (squares) observed associating with RAC substates dnDEL compared to WT (y-axis shows $-\log_{10}$ p-value) of enriched proteins (cutoff: L2FC > |0.7| and $p<0.01$); inset shows the number observed for those enriched in dnDEL (yellow) or WT (blue); schematic below shows model of RAC substates recruitment to distinct protein-associated species. (**G**) Western blot of NE (input) or WT, dnDEL (dn), or 2xDEL (2 x) RAC time-course (-ATP as described in E and F) for 7.5 min (left), 15 min (middle), or 30 min (right) probed with anti-panQki (top), anti-SF1 (middle), or anti-TATSF1 (bottom); MW shown in kD to the right.

The online version of this article includes the following source data for figure 4:

**Source data 1.** Original membranes corresponding to western blots depicted in *Figure 4D* (left), or *Figure 4G*.

**Source data 2.** Original membranes corresponding to western blots depicted in *Figure 4D* (left), or *Figure 4G*.

(2xDEL) mutants relative to WT; the proteome profiles associated with these two substates are also more similar to one another than the dnDEL comparison to WT, consistent with lower splicing efficiency or no splicing, respectively (*Figure 4E*). Closer and more stringent ($\log_2$ fold change $\geq$ |0.7| and $p<0.01$) inspection of the proteins binding preferentially to the dnDEL mutant RNA compared to the WT RNA reveals enrichment of 26 E/U2 components along with 16 other annotated (non-E/U2) RBPs; we observe depletion (or enrichment in WT) of only three E/U2 proteins but of 22 other annotated RBPs, including QKI (*Figure 4F*). Interestingly, some of the most enriched protein components associating with the dnDEL RNA are TATSF1, Sf3A1, PPIL4, Sf3A3, Sf3B4, DNAJC8, Sf3A2, Sf3B3, Sf3B1, and FUBP1 (*Figure 4F*), many of which are known to interact with the bs and SF1 or are members of the Sf3 complex, which promotes spliceosome maturation by displacing SF1 from the bp and then recruiting the U2 snRNP (*Martínez-Lumbreras et al., 2024*; *Ebersberger et al., 2023*; *Schmitzová et al., 2023*; *Zhan et al., 2024*). The most significantly enriched proteins associating with the WT RAC substrate are NONO, SFPQ, PSPC1, and QKI; the former three are found in paraspeckles that are formed by liquid-liquid phase separation (*Knott et al., 2016*), and QKI has recently been identified as a paraspeckle component protein (*Dyakov et al., 2024*). Western blot analysis validates the above findings indicating that QKI associates with high affinity to the WT but not dnDEL or 2xDEL RNA, and, in the absence of ATP, SF1 and TATSF1 associate to a greater degree with the dnDEL mutant than either of the other RAC substrates (*Figure 4G*). Therefore, both ACUAAC elements are required for QKI binding and repression of Rai14 exon 11 splicing; removing the downstream ACUAAC element causes SF1 binding, subsequent recruitment of the spliceosome A complex machinery, and derepression of RAI14 exon 11 splicing.

## Ectopic expression of QKI5 in *S. cerevisiae* is lethal and causes pre-mRNA splicing defects

Our results indicate that QKI and SF1 can directly compete for a subset of ACUAA bp sequences, so we hypothesized that expressing the QKI5 isoform in *S. cerevisiae* (where the bp sequence is nearly invariant UACUAAC and lacks *QKI*) would be lethal and cause defective splicing. To test this, we inserted a galactose-inducible cDNA encoding either EGFP, mutant K120A;R124A QKI5, or wild-type QKI5 into BY4741 at the *URA3* locus, and each grew normally on glucose-containing media where the transgene is repressed. Galactose induction is tolerated for the EGFP- and mutant QKI5-containing yeast but is lethal for the wild-type QKI5-containing strain (*Figure 5A*). Growth curve analysis indicates that after about 4 hr, the QKI5-expressing BY4741 cells cease proliferation (*Figure 5—figure supplement 1A*). Interestingly, by 24 hr most of these cells have a 'large-budded' morphology, which suggests a defect in cell division; EGFP- and mutant QKI5-expressing cells grow normally (*Figure 5—figure supplement 1A and B*).

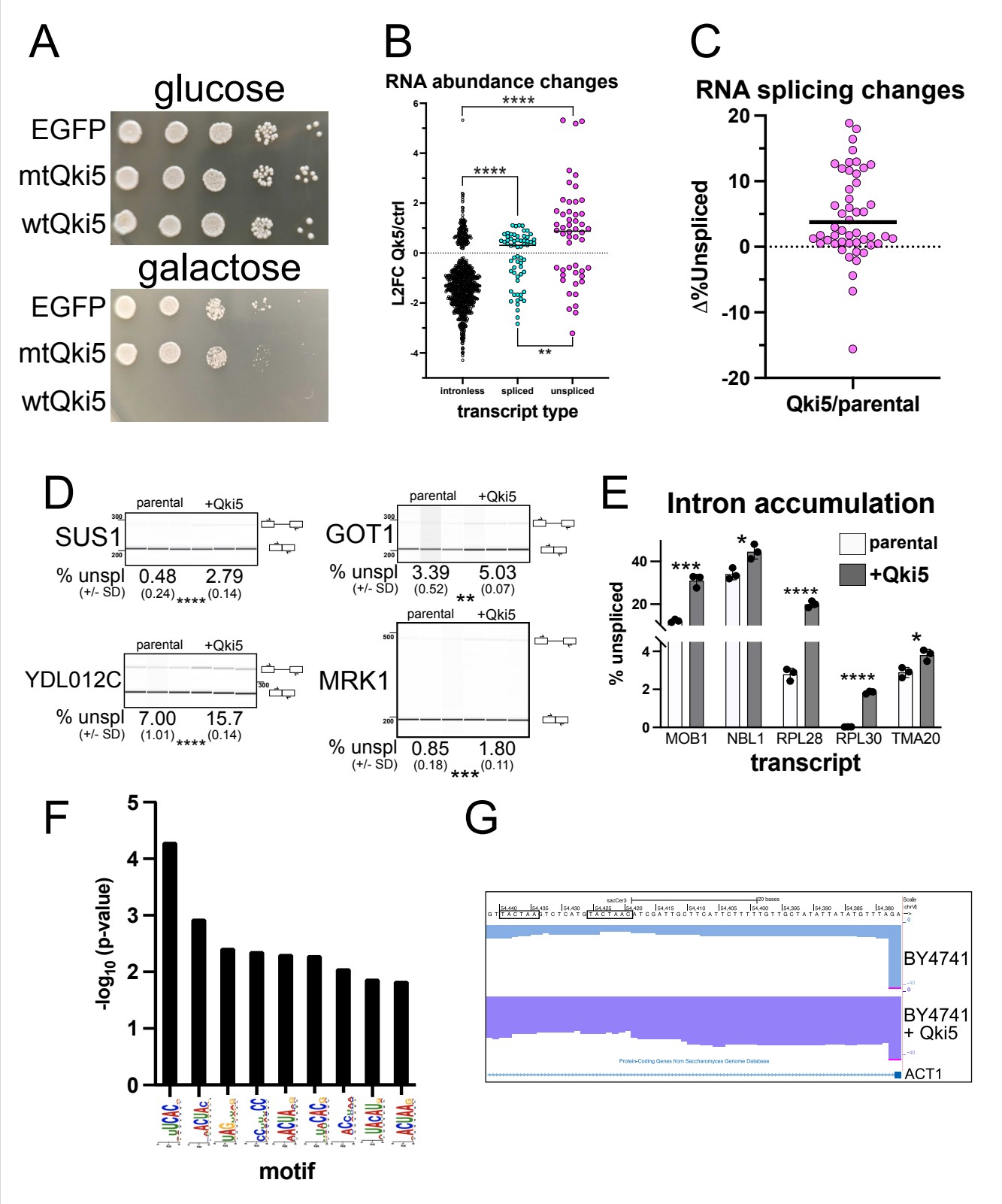

**Figure 5.** *QKI5* expression is lethal in yeast and represses splicing. (**A**) BY4741 *S. cerevisiae* strain with *EGFP* (top), mutant *QKI5* (mtQKI5; middle), or wild-type *QKI5* (wtQKI5; bottom) GAL-inducible transgene cultures grown on either glucose- (top) or galactose-containing (bottom) YPD plates at decreasing densities (left to right) and incubated at 30 °C for 72 hr. (**B**) Changes in RNA abundance measured by DESeq2 analysis of RNA-seq data for intronless, spliced, or unspliced transcripts were measured from RNA extracted from BY4741 cells with *QKI5* expression (induced by galactose-

*Figure 5 continued on next page*

*Figure 5 continued*

containing media for 4 hr), or in the BY4147 parental strain also cultured in galactose-containing media for 4 hr (control), and are shown as L2FC in *QKI5*-induced cells relative to control (n=3 biological replicates; cutoff *p*<0.1; abundance cutoff of TPM >0.2); **\*\****p*<0.01 or \*\*\*\**p*<0.0001. (**C**) Splicing changes measuring the change in percent unspliced (Δ%Unspliced; y-axis) for BY4741 as described in B.; cutoff *p*<0.1 by Student's t-test and base mean >100. (**D**) RT-PCR with primers that span exon-intron-exon junctions and BioAnalyzer gel-like image showing mean percent unspliced (+/- SD; n=3) below for introns whose inclusion increased upon *QKI5* ectopic expression and as measured by RNA-seq analysis in C. for the parental or *QKI5*-expressing cells (\*\**p*<0.01, \*\*\**p*<0.001, \*\*\*\**p*<0.0001 by Student's t-test); amplicon size in bp is shown to the left for each except YDL012C, which is shown to the right. (**E**) RT-qPCR analysis measuring mean percent unspliced transcript from RNA extracted from biological triplicate cultures of either the parental control or BY4741 expressing *QKI5* for each transcript shown (+/- SD; *\*p*<0.05, \*\*\**p*<0.001, \*\*\*\**p*<0.001). (**F**) Bar graph showing -log₁₀ *p*-values (y-axis) of significantly enriched (SEA; *p*<0.01) motifs observed in 3' proximal ends of introns whose inclusion increases upon ectopic *QKI5* expression in yeast (x-axis). (**G**) UCSC Genome Browser screenshot showing RNA-seq reads mapping to the *ACT1* transcript intron/exon junction near the 3'ss from RNA extracted from parental control and *QKI5*-induced cells; boxed sequences show two TACTAA elements.

The online version of this article includes the following source data and figure supplement(s) for figure 5:

**Source data 1.** Original gel-like images from BioAnalyzer used in *Figure 5D*.

**Source data 2.** Original gel-like images and data file from BioAnalyzer used in *Figure 5D*.

**Figure supplement 1.** QKI5 expression is lethal in yeast and represses splicing.

**Figure supplement 1—source data 1.** Original uncropped agarose gel images of data shown in *Figure 5—figure supplement 1C*.

**Figure supplement 1—source data 2.** Original uncropped agarose gel images of data shown in *Figure 5—figure supplement 1C*.

Next, we tested how ectopic QKI5 expression impacts the yeast transcriptome. The parental BY4741 strain or BY4741 with WT QKI5 was grown in galactose-containing media for 4 hr, and then RNA was collected and sequenced. We used a *S. cerevisiae* genome annotation that specifies intronic regions, pre-mRNAs, spliced mRNAs, and intronless transcripts (*Talkish et al., 2019*) to map RNA-seq reads to the transcriptome and then measured these transcript types. We observe higher levels of unspliced pre-mRNA and spliced mRNA when comparing the changes in QKI5 induction to the control, while intronless transcript abundance decreases (cutoff *p*<0.1; *Figure 5B* and *Supplementary file 6*). Unspliced pre-mRNAs accumulate to a greater degree than spliced mRNAs, supporting the notion that ectopic QKI5 expression perturbs splicing in yeast (\*\**p*<0.01 by Mann-Whitney U; *Figure 5B*). We measured changes in splicing by calculating the percentage of unspliced pre-mRNA for each intron-containing transcript (n=304 (out of 325 total) that passed expression level cutoff of base mean > 100) and discovered that 50 changed significantly (cutoff *p*<0.1 by Student's t-test; *Supplementary file 7*) upon QKI5 expression. Of these, 41 increase in percent unspliced (~12% of all expressed intron-containing transcripts) while nine decrease compared to control, which indicates a strong bias toward intron accumulation upon ectopic QKI5 expression (*Figure 5C*). We validated several of these using RT-PCR (*Figure 5D* n=4) and (*Figure 5—figure supplement 1C* n=15) or RT-qPCR (*Figure 5E* n=5) and found that each of these instances of repressed splicing upon QKI5 expression that is measured by RNA-seq is also observed using these complementary methods. Notably, the HAC1 intron, which is not removed by the spliceosome (*Sidrauski et al., 1996*), shows no increase in intron accumulation upon ectopic QKI5 expression by either RNA-seq or RT-PCR (*Figure 5—figure supplement 1C*). Next, we asked if any intron sequence motifs are enriched that correlate with increased intron retention upon QKI5 expression by obtaining 80 nt of intron sequence upstream of the 3'ss and compared these to 80 nt of intron sequence in 108 control introns that are expressed but unchanged compared to control using SEA (*Bailey and Grant, 2021* see Methods). Nine motifs are significantly enriched (*p*<0.01), four of which have varying degrees of resemblance to the QKI binding motif (ACUAA) or 'half-site' (UAAY) (*Galarneau and Richard, 2005*; *Ryder and Williamson, 2004*; *Hafner et al., 2010*). We note that several of these could potentially serve as tandem embedded motifs (for example, U*ACUAA*CUAAC, where the first is italicized and the second underlined). A search for these finds that 18% of the introns that accumulated upon QKI5 induction compared to control have either two independent or tandem embedded motifs. In contrast, examination of the introns that are unchanged upon QKI5 induction reveals that only 6% had two intron ACUAA motifs within 80 nt of the 3'ss. The occurrence of two ACUAA elements per intron 3'ss-proximal region is significantly overrepresented in the set of introns that accumulate upon QKI5 ectopic expression (*p*=0.035 by Chi-squared test), suggesting that these are more sensitive to QKI5 splicing repression. Intriguingly, the essential ACT1 pre-mRNA has dual ACUAA elements in its intron proximal to the 3'ss, like those observed in RAI14

intron 10 (*Figure 2A*), and its splicing is blocked by QKI5 induction (*Figure 5F*). Thus, QKI5 expression in yeast is lethal and is, at least in part, due to splicing inhibition.

## Discussion

Our study reveals that QKI can repress the inclusion of alternatively spliced exons by directly competing with SF1/BBP for ACUAA bs. We find that 43% of the alternatively spliced exons co-regulated by both SF1 and QKI fall within the 'QKI repressed and SF1 activated' category and are associated with an ACUAA sequence motif/bs (*Figures 1 and 2*). We show unambiguously that either of the two ACUAAC elements in RAI14 intron 10 can be used as a bs, but that one or the other is required for any splicing and so must be a bs (*Figure 3*). Interestingly, deleting either ACUAAC element reduces exon skipping (*Figure 3*) and QKI5 binding (*Figure 4*), but increases inclusion (*Figure 3*) and SF1 binding (*Figure 4*). Therefore, QKI binds with high affinity to a bona fide dual bs in RAI14 intron 10 to prevent SF1 binding and splicing activation. In budding yeast, where the bs is nearly invariant ACUAAC, ectopic expression of QKI5 blocks pre-mRNA splicing and is lethal (*Figure 5*). Together, these findings demonstrate the validity of the bp competition model by showing these two RBPs directly compete for ACUAA bs substrates.

### Consequences of bp competition for pre-mRNA substates and other RNAs

One of the best-studied examples of competition between RBPs for pre-mRNA substrates is the competition between SR proteins, which promote exon activation, and hnRNPs, which repress exon inclusion (*Mayeda and Krainer, 1992*; *Lin and Fu, 2007*). This is the predominant manner through which most cell/tissue types carry out alternative pre-mRNA splicing. The model for bp competition between SF1 and QKI that we define here describes a novel but relatively simple pathway through which a specific subset of alternatively spliced exons with ACUAA bs can be regulated. Our findings are consistent with the established role of SF1 in bp recognition and splicing activation and suggest that QKI is more often a splicing repressor. A nuanced implication of this model is that the loss of function of one of these proteins leads to a gain of function in the other, by relieving competitive inhibition of a specific number of ACUAA bs that control the inclusion of certain alternatively spliced exons.

How does competition between SF1 and QKI influence transcript fate? In the case of *RAI14* intron 10, the dual ACUAA bs constitute a high-affinity bipartite binding site for dimeric QKI5 protein. In C2C12 myoblasts where QKI5 levels are high, *Rai14* exon 11 is mostly skipped (*Figure 2*), and this correlates with QKI5 binding (*Figures 2 and 4*). This binding event appears to promote paraspeckle association, as the paraspeckle proteins NONO, SFPQ, and PSPC1 (*Knott et al., 2016*) co-associate with QKI5-bound *RAI14* intron 10 (*Figure 4F*). This finding is consistent with a recent report that identifies QKI as a paraspeckle-associated protein (*Dyakov et al., 2024*). It is unclear how paraspeckle localization might influence splicing outcomes, but pre-mRNA localization within the nuclear speckle is associated with more efficient splicing (*Bhat et al., 2024*). It is possible that QKI-directed paraspeckle localization is repressive to pre-mRNA splicing or promotes the ligation of the exons flanking an alternatively spliced one, leading to exon skipping, as we observe in *RAI14*. Further study is required to elucidate how competition between SF1 and QKI for pre-mRNA substates influences their subnuclear localization, splicing, and stability. A recent study suggests that this may be complicated by the fact that QKI itself attenuates paraspeckle biogenesis by promoting the expression of the short *NEAT1* lncRNA isoform, which has lower paraspeckle-promoting activity than the longer isoform (*Zakutansky et al., 2024*). It is unclear if SF1 might have a similar role, which is possible, given that QKI promotes expression of the short isoform of NEAT1 via ACUAA elements (to which SF1 might also bind). Therefore, this may constitute a complex feedback loop in which QKI (and/or SF1) can regulate splicing in a paraspeckle-dependent manner but also regulates the degree to which paraspeckles accumulate.

### Branch site competition defines a subset of cell-type-specific alternative splicing

Competition for alternatively spliced exons between SR proteins and hnRNPs largely explains the splicing patterns observed in most cell types, but not in the most divergent patterns observed in muscle or brain cells (*Merkin et al., 2012*; *Barbosa-Morais et al., 2012*). This is due in part to

muscle- or brain-specific RBP expression patterns, and our study uncovers the novel finding that bp competition between SF1 and QKI helps explain how these are enforced. SF1 levels are relatively high in most tissues but can vary by up to eightfold (*Tapial et al., 2017*), while QKI levels are much more dynamic. QKI levels are elevated in muscle, but undetectable in most neurons or predominated by the cytoplasmic isoforms QKI6 and QKI7 in glia (*Noveroske et al., 2002*; *Hardy et al., 1996*; *Ebersole et al., 1996*; *Kondo et al., 1999*; *Feng et al., 2021*). Therefore, in muscle, the nuclear QKI:SF1 ratio is high, whereas in many brain cell types, it is low (or zero), resulting in opposite patterns of isoform-specific transcriptomes and proteomes within the context of the bs competition model. This is also relevant in developmentally regulated systems, where QKI5 levels increase during cardiac cell differentiation (*Mazin et al., 2021*) but decrease as neural stem cells commit to mature cell types (*Hayakawa-Yano et al., 2017*). Similarly, when embryonic stem cells exit pluripotency and commit to the endoderm lineage, QKI levels decrease; however, upon commitment to the mesodermal lineage, QKI levels increase, and a QKI-associated splicing gain-of-function is observed (*Fagg et al., 2022*). On the other hand, SF1 reduction, leading to increased unspliced RNA and skipped exons, has been observed in a model of aging and correlates with age-related fitness decline (*Heintz et al., 2017*). In summary, the competition between SF1 and QKI appears to be a novel molecular mechanism through which different cell-type-, tissue-, lineage-, and developmental-specific, as well as homeostatic, splicing programs can be achieved and regulated.

## Evolutionary implications of bp competition

Yeast and other single-celled eukaryotes lack *QKI*, whereas multicellular eukaryotes and plants have the gene or an ortholog. Strikingly, we found that 12% of the introns expressed in yeast strain BY4741's splicing was inhibited by ectopic QKI5 expression, and that these cells developed a large-budded phenotype due to a failure to divide (*Figure 5—figure supplement 1A and B*). This phenotype has been observed due to a lack of *TUB1* pre-mRNA splicing (*Burns et al., 2002*), although we did not observe mis-splicing of it (or *TUB3*) in our study. The introns most sensitive to QKI5 have evidence of the 'bipartite' QKI motif, which is consistent with high-affinity binding of dimeric QKI (*Galarneau and Richard, 2005*) and similar to that observed in RAI14 intron 10. It is interesting that yeast lack *QKI*, do not exhibit 'alternative splicing' per se, and have a more invariant bs. This is UACUAAC in *S. cerevisiae* (*Berglund et al., 1997*; *Moore et al., 1993*; *Gould et al., 2016*), but *Schizosaccharomyces pombe* has a slightly more degenerate bs (*Kupfer et al., 2004*), and its introns share some features with both budding yeast and mammals (*Prabhala et al., 1992*; *Käufer and Potashkin, 2000*). Nevertheless, the consensus bs in the latter still represents a strong QKI-binding motif (*Galarneau and Richard, 2005*; *Ryder and Williamson, 2004*), suggesting that QKI expression might also be lethal. Thus, it would be evolutionarily unfavorable to select for *QKI* or an ortholog/close relative.

The metazoan STAR family of RBPs, with the exception of SF1/BBP, which is the most divergent member, all contain the QUA1 dimerization domain (*Vernet and Artzt, 1997*). In the case of QKI (or ASD2 in *C. elegans Ohno et al., 2008*) or HOW in *Drosophila* (*Zaffran et al., 1997*), the binding motif ACUAA, along with the presence of a YAAY 'half site' within about 20 nt, requires more specificity and generates additional points of contact on an RNA, which may explain in part its higher binding affinity than SF1. Perhaps it would not have been evolutionarily advantageous for the majority of bs in multicellular organisms to be UACUAAC/high-affinity QKI motifs (fewer than 20% of human introns have this bs *Irimia and Roy, 2008*). So more degenerate bs were selected for in metazoans. However, the presence of some ACUAA bs could help diversify the functions of cell-type-specific protein isoforms. For example, many cytoskeletal/contraction-related protein isoforms are specific to muscle (*Nakka et al., 2018*; *Giudice et al., 2016*), and many of these are alternatively spliced exons that are repressed by QKI through binding UACUAAY in introns upstream of them (*Fagg et al., 2017*; *Sugnet et al., 2006*; *Hall et al., 2013*; *Mazin et al., 2021*). Similar observations in worms and flies suggest the intriguing possibility that this may have coevolved with the QKI gene and the more extensive bs degeneracy observed in metazoans and plants. One way this would have been executed, but also constrained, is through bs competition between QKI and SF1/BBP. Consequently, this may have enabled expansion of the transcriptome and proteome through diversification of alternative splicing.

# Materials and methods

**Key resources table**

| Reagent type (species) or resource | Designation | Source or reference | Identifiers | Additional information |
|---|---|---|---|---|
| Cell line (yeast) | BY4741 | Ares Lab/Prakash Lab | NCBI Taxonomy ID: 1247190 | |
| Cell line (mouse) | C2C12 | ATCC | CRL-1772 | |
| Cell line (human) | Flp-In T-Rex 293; WT or QKI KO | Thermo Fisher | RRID:CVCL_U427 | |
| Transfected construct (human) | pDUP-RAI14 (WT and various mutants) | Parental plasmid *Dominski and Kole, 1991* | | |
| Transfected construct (mouse) | pMyc:Qk5 | Sean Ryder | 'Wild-type' but contains a translationally silent mutation in the region corresponding to exon 6 | |
| Transfected construct (mouse) | pMyc:Qk5 (K120A;R124A) | *Fagg et al., 2022* | QKI5 defective in RNA binding | |
| Transfected construct (human) | pcDNA3.1-SF1 | This paper | | See '*Plasmids and Transfections*' section in Methods below |
| Other | pcDNA3.1-tdTomato | *Fagg et al., 2017* | | *Fagg et al., 2017* |
| Other | pJW1666 | Addgene plasmid # 112040 | | Jonathan Weissman (Addgene plasmid # 112040) |
| Other | pJW1666-Qki5 | Addgene plasmid # 112040 with Qki5 inserted | | Jonathan Weissman (Addgene plasmid # 112040) with Qki5 inserted |
| Other | pJW1666-Qki5 (K120A;R124A) | Addgene plasmid # 112040 with Qki5 (K120A;R124A) inserted | | Jonathan Weissman (Addgene plasmid # 112040) with Qki5 (K120A;R124A) inserted |
| Recombinant DNA reagent | pcDNA5-aptamer plasmid | Garcia-Blanco lab | | |
| Antibody | GAPDH; Mouse monoclonal IgM | Sigma | G8795 | 1:40,000 |
| Antibody | α/β-Tubulin; Rabbit polyclonal | Cell Signalling | 2148 | 1:5000 |
| Antibody | panQKI; Mouse monoclonal IgG2b (clone N147/6) | Sigma | MABN624 | 1:2000 |
| Antibody | SF1; Rabbit polyclonal | Bethyl | A303-213a | 1:5000 |
| Antibody | TATSF1; Rabbit polyclonal | Thermofisher | 20805–1-AP | 1:2000 |
| Antibody | IRDye 800CW Goat anti-Mouse IgG2b; Secondary anti-mouse | Licor | P/N 926–32352 | 1:15,000 |
| Antibody | IRDye 800CW Goat anti-Rabbit IgG Secondary anti-rabbit | Licor | P/N: 926–32211 | 1:15,000 |
| Antibody | IRDye 680RD Goat anti-Mouse IgM Secondary anti-mouse | Licor | P/N: 926–68180 | 1:20,000 |
| Antibody | IRDye 680RD Goat anti-Rabbit IgG Secondary anti-rabbit | Licor | P/N: 926–68071 | 1:20,000 |
| Commercial assay or kit | SYBR Green Universal Master Mix | Applied Biosystems | | |
| Commercial assay or kit | Taq2x Master Mix | New England Biolabs | | |

*Continued on next page*

*Continued*

| Reagent type (species) or resource | Designation | Source or reference | Identifiers | Additional information |
|---|---|---|---|---|
| Commercial assay or kit | ReliaPrep RNA mini prep kit | Promega | | |
| Commercial assay or kit | iScript Reverse Transcription Supermix | BioRad | | |
| Commercial assay or kit | NEBNext poly(A) mRNA Magnetic Isolation module | New England Biolabs | NEB, E7490 | |
| Commercial assay or kit | NEBNext Ultra II Directional RNA Library Prep kit for Illumina | New England Biolabs | NEB, E7760 | |
| Commercial assay or kit | HiScribe T7 High Yield RNA Kit | New England Biolabs | NEB, E2040 | |
| Software, algorithm | UCSF ChimeraX | https://www.rbvi.ucsf.edu/chimerax | RRID:SCR_015872 | Version 1.10.1 |
| Software, algorithm | Vast-tools | *Irimia et al., 2014*; *Tapial et al., 2017* | | 2.0.2, database has16.02.18 |
| Software, algorithm | rMATS | *Shen et al., 2014*; *Park et al., 2016* | | Version 4.0.2, GENCODE v30 annotation |
| Software, algorithm | rMAPS2 | rMAPS2; https://rmaps.cecsresearch.org/ | | |
| Software, algorithm | Trimmomatic | Trimmomatic | RRID:SCR_011848 | v.0.39 |
| Software, algorithm | Kallisto | Kallisto | RRID:SCR_016582 | v.0.50 |
| Software, algorithm | Deseq2 | *Love et al., 2023* | | v1.42.1 |
| Software, algorithm | Simple Enrichment Analysis (SEA) | *Bailey and Grant, 2021*; https://meme-suite.org/meme/doc/sea.html | | |
| Software, algorithm | XCalibur | Thermo Scientific | RRID:SCR_014593 | V2.5 |
| Software, algorithm | Proteome Discoverer | Thermo Fisher | RRID:SCR_014477 | Version 2.2.0388 |
| Software, algorithm | Scaffold | Proteome Software Inc. | RRID:SCR_014321 | Version Scaffold_4.11.1 |
| Software, algorithm | Protein Prophet | *Nesvizhskii et al., 2003* | | |
| Software, algorithm | MSConvert | *Chambers et al., 2012* | | |
| Software, algorithm | MSFragger | *Kong et al., 2017* | | |
| Software, algorithm | DIA-NN | https://github.com/vdemichev/DiaNN; *Demichev et al., 2020* | | |
| Software, algorithm | Fragpipe-Analyst | *Hsiao et al., 2024* | | |
| Sequence-based reagent | Oligonucleotides and RNAi | See *Supplementary file 8* | | |

## Cell culture

C2C12 mouse myoblasts (obtained from ATCC) and HEK293 cells were cultured in Dulbecco's Modified Eagle Medium (DMEM) supplemented with high glucose (Invitrogen) and 10% (v/v) heat-inactivated fetal bovine serum (Thermo Fisher). Cells were maintained at 37 °C in a humidified atmosphere with 5% $CO_2$. The HEK293 QKI KO cells were shared by Kuo-Chieh Liao and generated as previously described (*Liao et al., 2021*; *Liao et al., 2020*) and used the Flp-In T-Rex 293 cell line. The cells were routinely tested for mycoplasma and tested negative. STR phenotyping was conducted at the UTMB Molecular Genomics Core Facility on QKI KO and control HEK cells, confirming their identity as HEK293T cells.

## Protein alignment

The Quaking protein used for alignment is as referenced (*Teplova et al., 2013*) and used PDB accession 4JVH and for SF1 *Liu et al., 2001* used PDB 1K1G. Alignment and visualization were performed using UCSF ChimeraX (https://www.rbvi.ucsf.edu/chimerax) version 1.10.1 (*Meng et al., 2023*).

## Plasmids and transfections

The pDUP51 splicing reporter plasmid (*Dominski and Kole, 1991*) was used as a backbone to generate the RAI14 exon 11 plasmid. A 243 bp fragment upstream of exon 11, exon 11, and a 95 bp fragment downstream of the exon were PCR-amplified from H9 human embryonic stem cell genomic DNA; the forward primer contained an ApaI site, and the reverse primer contained a BglII site. These and the pDUP51 plasmid were digested with ApaI and BglII (NEB), then ligated into pDUP51 at the ApaI and BglII restriction sites. The resulting pDUP-RAI14 exon 11 plasmid was confirmed by Sanger sequencing. Mutant constructs were generated using the Q5 Site-Directed Mutagenesis Kit and were also verified by Sanger sequencing. The myc:QKI5 plasmids used were described previously and were a kind gift from Sean Ryder (*Fagg et al., 2017*; *Fagg et al., 2022*). Generation of the pcDNA3.1-tdTomato plasmid was also previously described (*Fagg et al., 2017*). The pcDNA3.1-SF1 plasmid was generated by Gibson Assembly using cDNA from H9 human embryonic stem cells as a template for PCR amplification and confirmed by Sanger sequencing.

Transfections were carried out with Lipofectamine 2000 (Invitrogen) using an 'in tube transfection' protocol, where $2.5 \times 10^5$ cells were added to a tube containing Lipofectamine and the appropriate volume of reagent, following the manufacturer's instructions, with 100 ng DNA mix in Gibco Opti-MEM (Thermo) or 30 pmol of siRNA, incubated for 20 min at RT, plated in 12 well plates containing DMEM 10% FBS and incubated overnight. Cells were harvested 24 hr after transfection.

For the RNA affinity chromatography, the j6f1 aptamer under the control of the T7 RNA promoter was obtained by the digestion of the vector pcDNA5-aptamer plasmid and was a gift from Chloe Nagasawa (University of Texas Medical Branch, Galveston, Texas, US) from the lab of Mariano Garcia-Blanco (University of Virginia). RAI14 sequence consisting of 60 bp fragment of RAI14 intron 10 upstream RAI14 exon 11 was obtained from a gBlock (IDT), digested with HindIII and Not1 restriction enzymes and cloned into the digested PCNA5 aptamer plasmid.

For yeast transformation, the plasmid pJW1666 (gift from Jonathan Weissman Addgene plasmid # 112040; http://n2t.net/addgene:112040; RRID:Addgene_112040; *Costa et al., 2018*) was used to clone the WT or mutant QKI sequences, or the GFP present on WT the plasmid was used as control.

## RNA extraction, RT-PCR, and RT-qPCR

RNA was extracted from cells using Trizol (Thermo Fisher). For direct extraction from cell plates, Trizol was added to the wells and samples were vortexed. For extraction from cell lysates, cells were extracted with RSB100 (100 mM Tris-HCl pH 7.4, 0.5% (v/v) NP-40, 0.5% (v/v) Triton X-100, 0.1% (w/v) SDS, 100 mM NaCl, and 2.5 mM MgCl$_2$). Samples were vortexed, chloroform was added, then the samples were incubated at RT for 2 min. After centrifugation at 13,000 RPM for 15 min at 4 °C, the aqueous phase was transferred to a new tube and samples were processed with the ReliaPrep RNA MiniPrep system (Promega), according to the manufacturer's instructions. Reverse transcription was performed with iScript Reverse Transcription Supermix (BioRad) according to manufacturer instructions. RT-PCR was performed using Taq2x Master Mix (NEB), and cycle numbers were determined empirically to prevent over-amplification. PCR products were analyzed by agarose gel and via Agilent 2100 BioAnalyzer.

For RT-qPCR, cDNA was diluted 1:8 and used with Applied Biosystems SYBR Green Universal Master Mix on a Step One Plus Real-Time PCR machine (Applied Biosystems). RT-qPCR cycling conditions were: 95 °C for 10 min, [95 °C 15 s, 60 °C 60 s] (40 cycles). DUP-RAI14 exon 11 reporter RNA abundances were analyzed by calculating the ΔCt values normalized to EEF1A1. The ΔΔCt values were then calculated as 2^-ΔCt for each sample. The average ΔΔCt of the wild-type reporter was used as a normalization factor to determine the relative RNA abundance in the mutant reporters.

## Western blotting

Proteins were extracted using RSB100 (100 mM Tris-HCl pH 7.4, 0.5% (v/v) NP-40, 0.5% (v/v) Triton X-100, 0.1% (w/v) SDS, 100 mM NaCl, and 2.5 mM MgCl$_2$) with EDTA-free protease inhibitor cocktail

(SIGMAFAST, Sigma) and 1 mM PMSF. The buffer was added directly to the wells of the plates, scraped, collected, and kept on ice, while vortexing every 5 min for 30 min, followed by centrifugation at 14,000 RPM for 15 min at 4 °C. The supernatant was transferred to a new tube. Samples were stored at –80 °C until use.

Protein concentration was measured using the Bradford assay (BioRad *Bradford, 1976*). The same amount of protein from each sample was loaded onto 10% SDS-PAGE (typically 15–30 µg) and then transferred to a 0.45 µm nitrocellulose membrane (Thermo). The membrane was blocked (with 5% non-fat dry milk dissolved in tris-buffered saline TBS) for 1 hr at room temperature with constant agitation. Primary antibody incubations were performed overnight at 4 °C with constant agitation in TBS 0.01% (v/v) tween (TBS-T) with 5% non-fat dry milk using the following antibodies: Anti-Pan-QKI Antibody, clone N147/6 (MABN624, Sigma-Aldrich) (1:2000), IgM-anti-GAPDH (G8795, clone GAPDH-71.1, Sigma-Aldrich) (1:40000), α/β-Tubulin (#2148, Cell Signaling) (1:5000), anti-SF1 (#A303-213a, Bethyl Laboratories) (1:5000), and anti-HTATSF1 (# 20805–1-AP, Thermo Fisher) (1:2000), all diluted in TBS containing 5% (w/v) milk, and 0.01% (v/v) Tween-20 (TBST). The following day, the membranes were washed three times with TBST with 5% (w/v) milk at RT. Secondary infrared conjugated antibodies: IRDye 800CW Goat anti-Mouse IgG2b (P/N 926–32352), IRDye 680RD Goat anti-Mouse IgM (P/N: 926–68180), IRDye 800CW Goat anti-Rabbit IgG (P/N: 926–32211), and IRDye 680RD Goat anti-Rabbit IgG (P/N: 926–68071) from Li-Cor were diluted according to manufacturer instructions (1:15,000 for 800CW or 1:20,000 for 680RD/LT conjugates) in TBST with 5% (w/v) non-fat dry milk for 1 hr with constant agitation. Membranes were washed three times every 5 min before visualization on the Odyssey CLx imager (Li-Cor).

## Yeast transformation

Yeast transformation was performed as described by Ito et al. with modifications (*Ito et al., 1983*) using the BY4741 strain. A 50 mL culture of BY4741 was grown to a density of $5 \times 10^6$ overnight. Following centrifugation, the pellet was resuspended in 1 ml of deionized water, centrifuged at 12,000 RPM and resuspended in 0.5 ml of 1 X TE-LiAc (100 mM tris-HCl, 10 mM EDTA pH 7.5; LiAc pH 7.5). Subsequently, cells were resuspended in 3 x volume of 1 X TE-LiAc and incubated at 30 °C with agitation. 2 µg of purified PCR of WT QKI, mutant QKI and GFP plasmids, and 7 µL of salmon sperm carrier DNA were added to 200 µL of the yeast cells and the mixture was incubated for 45 min at 30 °C with agitation. The following steps included a 3 hr incubation at 30 °C with agitation, a 20 min heat shock at 42 °C, centrifugation at 12,000 RPM, and resuspension in YEPD followed by a 45 min incubation. After a 12,000 RPM centrifugation, cells were resuspended in water and spread in URA- selection plates, then incubated at 30 °C until transformants appeared.

## Yeast genomic DNA extraction

Genomic DNA extraction was performed using the method developed by Klassen and collaborators with a few modifications (*Klassen et al., 2008*). Cells were patched on URA- selection plates, and 50 µL of cell volume was scraped off the plate and resuspended in zymolase 20T 8 mg/ml. After vortexing the samples, they were incubated at 37 °C for 1 hr. After incubation, the cells were spun down, the supernatants were aspirated, and the cells were resuspended in 250 µL of 10% SDS. Then, they were vortexed and incubated for 30 min at 65 °C. After incubating the samples on ice for 5 min, 100 µL of 5 M potassium acetate, pH 7.5, was added, and the samples were incubated on ice for 1 hrr. Tubes were spun at 14,000 RPM for 15 min, and the supernatant was transferred to a new tube. Then 400 µL of ice-cold 95% EtOH was added, vortexed briefly, and spun for 15 min at 14,000 RPM. The supernatants were decanted, and the DNA pellet was dried at 30 °C for 15 min. It was then dissolved in 100 µL of ultra-pure water and placed in a mixer for 20 min at 2000 RPM. The extracted genomic DNA was used to perform PCR with high-fidelity polymerase (Takara Bio Inc) to confirm the transgene expression by Sanger sequencing.

## Yeast RNA extraction

RNA was extracted from yeast cells as previously described (*Rio, 2011*). Briefly, 1 mL of yeast culture was pelleted and then resuspended in 400 µL of AE buffer (50 mM, pH 5.2 of NaOAc, 10 Mm EDTA). After the addition of 40 µL of 10% SDS and 400 µL of PCA, the samples were incubated at 65 °C for 10 min. After a 5 min incubation on ice, samples were placed in phase lock gel tubes, centrifuged

at 14,000 RPM for 5 min, and centrifuged again using the same conditions following another chloroform addition; this chloroform wash was repeated, and the samples were centrifuged one final time. The aqueous phase was transferred to a new 1.5 mL tube, 50 µL of 3 M sodium acetate, pH 5.2, was added, followed by two volumes of 100% ethanol. Samples were centrifuged at 14,000 RPM for 15 min, the ethanol was removed, and the pellet was washed with 70% ethanol. Then, the samples were centrifuged at 14,000 RPM for 5 min, the ethanol was removed, and the samples were air-dried. The pellet was resuspended in RNAse-free water and then vortexed. To remove any potential DNA contamination, 1 µg of RNA was treated with DNase Turbo following the manufacturer's instructions. RNA was extracted after DNase treatment with phenol/chloroform/isoamyl alcohol and then ethanol precipitation. RNA concentration was measured using the NanoDrop (Thermo Fisher).

## Yeast spot assay

One milliliter of freshly harvested cells grown overnight in 10 mL of YEPD were centrifuged, washed in ultra-pure water and sonicated. The cells were then diluted to a concentration of $1 \times 10^7$. Serial 10-fold (up to five times) were prepared on both galactose (Gal) and non-galactose plates. The plates were incubated at 30 °C for 3 days.

## Yeast growth curve

Yeast was grown in YEPD overnight. When the OD660 reached approximately 0.5, transgene activation was induced by adding galactose dissolved in water to a final concentration of 2% or water as control. For each time point, samples were collected for cell count and RNA extraction.

## RNA sequencing and analysis

RNA-seq data from the ENCODE project for shQKI and shNT (GSE80878) or shSF1 and shNT (GSE88430) (*Van Nostrand et al., 2020a*; *The ENCODE Project Consortium, 2012*) were downloaded from GEO and analyzed as previously described (*Fagg et al., 2022*) using Vast-tools (*Irimia et al., 2014*; *Tapial et al., 2017*) or rMATS (*Shen et al., 2014*) to compare alternative splicing observed in shQKI relative to shNT, or shSF1 relative to shNT.

The subsequent Vast-tools data files were filtered using a cutoff of dPSI > |10| and MVdPSI95>0 to identify alternatively spliced events that changed significantly upon either QKI or SF1 knockdown relative to the control. For 'co-regulated' exons, the above cutoff was required for either shQKI compared to control or shSF1 compared to control, but in both datasets, the MVdPSI95 had to be >0. rMATS analysis and rMAPS2 (*Park et al., 2016*; *Hwang et al., 2020*) motif enrichments were done using default conditions and with custom motifs added to measure various bp-like and potential SF1 binding motifs (ACT[ACTG]AG, [ACTG]CT[AG][CT], TAA[CT], TAA[CT]T[ACTG]A[CT], TACTAAC, TACTAA, ACTAA[CT], TACTAA[CT], CTAAC[ACG]).

Strand-specific RNAseq libraries were prepared from 1 µg of yeast total RNA using NEBNext poly(A) mRNA Magnetic Isolation module (NEB, E7490) and NEBNext Ultra II Directional RNA Library Prep kit for Illumina (NEB, E7760) following the manufacturer's recommended procedure. The six libraries were pooled in equal molar concentration and sequenced on Illumina NextSeq 550 for PE 150 base pair sequencing, yielding about 30 M paired-end reads each. The associated data files have been uploaded to GEO under accession GSE273838. The RNAseq reads were filtered for low-quality bases and trimmed of adapter sequences using Trimmomatic (v.0.39). The trimmed reads were aligned to the yeast reference genome SacCer3 and a custom yeast annotation file, which accounts for all yeast introns (*Hardy et al., 1996*) (and allows measurement of unspliced, spliced, and intronless transcripts) using kallisto (v.0.50) to generate the abundance.tsv file for either normalized (TPM) or total (counts) reads for each sample. The TPM file was used as input for DESeq2 analysis (v1.42.1) to identify changes in transcript abundance (abundance cutoff TPM > 0.2 and significance cutoff $p>0.1$; *Supplementary file 6*), and the read counts file was used to calculate the percentage of unspliced RNA (percent unspliced = (unspliced/(unspliced+spliced))*100) for each transcript that was expressed (base mean > 100) and statistical significance was measured using the Student's t-test with a change in percent unspliced $p<0.1$ considered significant (*Supplementary file 7*).

Motif analysis of both the yeast and HepG2 RNA-seq datasets were performed using SEA (*Bailey and Grant, 2021*), and the former used 80 nt of intron sequence upstream of the 3'ss of yeast introns, 38 of which increased ($p<0.1$) upon ectopic QKI5 expression compared to the parental control (this

represents 38 of the 41 we observed that increased in inclusion upon QKI5 expression, as the other two were located on ChrM and did not overlap with 'Talkish Standard Introns' *Talkish et al., 2019*, which was a requirement for our motif analysis), compared to a background set of 106 introns that were detectable (base mean > 100) but unchanged ($p$>0.2, change in percent unspliced < |1|) upon ectopic QKI5 expression compared to the parental control. For the HepG2 intron set, we extracted the intron sequence that began 20 nt upstream of the 3'ss (to exclude analyzing potential differences in 3'ss or pY tracts) and spanned 60 nt upstream of the 3'ss of introns that were co-regulated by QKI and SF1 (dPSI > |10| and MVdPSI at 95% confidence interval >0 in either dataset or the other and at least MVdPSI at 95% confidence interval >0 in both datasets, as determined by Vast-tools) and performed SEA, and then also performed an identical analysis but using 1000 introns that were detectable (base mean >100) but unchanged in either shQKI relative to shNT or shSF1 relative to shNT (dPSI < |1|, MVdPSI = 0). In both cases, the motif set that was interrogated was from *Ray et al., 2013*.

## RNA affinity chromatography

Tobramycin RNA affinity chromatography was performed as previously described (*Hartmuth et al., 2002*) with several minor modifications. Briefly, the aptamer and RAI14 WT and mutant RNAs were produced after the vectors were digested with NotI (NEB) using the HiScribe T7 High Yield RNA Kit (NEB). Then 120 picomoles of RNA of the aptamer and RAI14 mutants, and 200 picomoles of wt RAI14 were heated in RNA binding buffer (20 mM Tris-HCl, 1 mM CaCl2, 1 mM MgCl2, 300 mM KCl, 0.1 mg/ml tRNA, 0.5 mg/ml BSA, 0.01% (v/v) NP-40, 0.2 mM DTT) at 95 °C for 5 min and transferred to RT for 30 min. Samples were incubated with 60 µL of tobramycin-coupled Sepharose matrix at 4 °C for 2 hr with head-over-tail rotation. The beads were washed three times with washing buffer (20 mM Tris-HCl, 1 mM CaCl2, 1 mM MgCl2, 145 mM KCl, 0.1% (v/v) NP-40, 0.2 mM DTT) and incubated with a 285 µL solution of 32% of C2C12 NE (as previously described *Dignam et al., 1983*), 32 mM KCL, 2 mM MgCl2, 2 mM ATP, and 20 mM creatine phosphate (CP) (the reactions were also performed without MgCl2, ATP and CP in the NE solution) for 7.5, 15, and 30 min at 30 °C with head over tail rotation. After NE incubation, the matrix was washed three times with a higher salt concentration of washing buffer (150 mM KCl). The beads were eluted with 5 mM of tobramycin in 20 mM Tris-HCl, 1 mM CaCl2, 1 mM MgCl2, 145 mM KCl, 2 mM MgCl2, 0.2 mM DTT in a 125 µL solution. The protein eluate was precipitated with acetone and suspended in LDS sample buffer for WB and in 5% (w/v) SDS, 50 mM TEAB pH 7.1 for MS analysis.

## Protein precipitation

Acetone was used to precipitate the proteins from the RNA affinity chromatography-eluted samples. Four times the sample volume of cold acetone was added to each sample and vortexed. Samples were incubated for 60 min at –20 °C for 1 hr, following a 10 min centrifugation at 14,000 RPM at 4° C. The acetone was decanted, and the samples were air-dried for 15 min. Pellets were resuspended with 1 x Laemmli buffer for western blot or with 5% (w/v) SDS, 50 mM TEAB, pH 7.1, for liquid chromatography with tandem mass spectrometry.

## Protein digestion

The samples were prepared as previously described (*Herrmann et al., 2021*). Briefly, 25 µg of protein from the above were reduced with 10 mM Tris(2-carboxyethyl) phosphine (TCEP) (77720, Thermo) and incubated at 65 °C for 10 min. The sample was then cooled to room temperature, and 1 µL of 500 mM iodoacetamide acid was added and allowed to react for 30 min in the dark. Then, 3.3 µL of 12% phosphoric acid was added to the protein solution followed by 200 µL of binding buffer (90% Methanol, 100 mM TEAB pH 8.5). The resulting solution was added to the S-Trap spin column (https://protifi.com/) and passed through the column using a bench-top centrifuge (60 s spin at 1000×*g*). The spin column is washed with 150 µL of binding buffer and centrifuged. This is repeated twice. 30 µL of 20 ng/µL trypsin is added to the protein mixture in 50 mM TEAB pH 8.5 and incubated at 37°C overnight. Peptides were eluted twice with 75 µL of 50% acetonitrile, 0.1% (v/v) formic acid. Aliquots of 20 µL of eluted peptides were quantified using the Quantitative Fluorometric Peptide Assay (Pierce, Thermo Fisher Scientific). Eluted volume of peptides corresponding to 5.5 µg of peptides are dried in a speed vac and resuspended in 27.5 µL 1.67% (v/v) acetonitrile, 0.08% (v/v) formic acid, 0.83% (v/v) acetic acid, 97.42% water, and placed in an autosampler vial.

## Data-dependent acquisition NanoLC MS/MS analysis

Peptide mixtures were analyzed by nanoflow liquid chromatography-tandem mass spectrometry (nanoLC-MS/MS) using a nano-LC chromatography system (UltiMate 3000 RSLCnano, Dionex), coupled on-line to a Thermo Orbitrap Fusion mass spectrometer (Thermo Fisher Scientific, San Jose, CA) through a nanospray ion source (Thermo Scientific) similar to as we have described previously (*Liu et al., 2022*). A trap and elute method was used. The trap column was a C18 PepMap100 (300 µm×5 mm, 5 µm particle size) from Thermo Scientific. The analytical column was an Acclaim PepMap 100 (75 µm×25 cm) from (Thermo Scientific). After equilibrating the column in 98% solvent A (0.1% formic acid in water) and 2% solvent B (0.1% (v/v) formic acid in acetonitrile (ACN)), the samples (2 µL in solvent A) were injected onto the trap column and subsequently eluted (400 nL/min) by gradient elution onto the C18 column as follows: isocratic at 2% B, 0–5 min; 2 to 24% B, 5–86 min; 24 to 44% B, 86–93 min; 44 to 90% B, 93–95 min; 90% B for 1 min, 90 to 10% B, 96–98 min; 10% B for 1 min 10 to 90% B, 99–102 min 90 to 4% B; 90% B for 3 min; 90 to 2%, 105–107 min; and isocratic at 2% B, till 120 min.

All LC-MS/MS data were acquired using XCalibur, version 2.5 (Thermo Fisher Scientific) in positive ion mode using a top speed data-dependent acquisition (DDA) method with a 3 s cycle time. The survey scans (*m/z* 350–1500) were acquired in the Orbitrap at 120,000 resolution (at *m/z*=400) in profile mode, with a maximum injection time of 100 ms and an AGC target of 400,000 ions. The S-lens RF level is set to 60. Isolation is performed in the quadrupole with a 1.6 Da isolation window, and CID MS/MS acquisition is performed in profile mode using rapid scan rate with detection in the ion-trap, with the following settings: parent threshold = 5000; collision energy = 32%; maximum injection time 56 ms; AGC target 500,000 ions. Monoisotopic precursor selection (MIPS) and charge state filtering were on, with charge states 2–6 included. Dynamic exclusion is used to remove selected precursor ions, with a +/-10 ppm mass tolerance, for 15 s after acquisition of one MS/MS spectrum.

## DDA database searching

Tandem mass spectra were extracted and charge state deconvoluted by Proteome Discoverer (Thermo Fisher, version 2.2.0388). Charge state deconvolution and deisotoping were not performed. All MS/MS samples were analyzed using Sequest (Thermo Fisher Scientific, San Jose, CA, USA; in Proteome Discoverer 2.5.0.402). Sequest was set up to search the canonical mouse proteome and a contaminant database cRAP, assuming the digestion enzyme trypsin. Sequest was searched with a fragment ion mass tolerance of 0.60 Da and a parent ion tolerance of 10.0 PPM. Carbamidomethyl of cysteine was specified in Sequest as a fixed modification. Deamidated of asparagine and glutamine, oxidation of methionine and acetyl of the N-terminus were specified in Sequest as variable modifications.

## Criteria for protein identification

Scaffold (version Scaffold_4.11.1, Proteome Software Inc, Portland, OR) was used to validate MS/MS-based peptide and protein identifications. Peptide identifications were accepted if they could be established at greater than 95.0% probability by the Scaffold Local FDR algorithm. Protein identifications were accepted if they could be established at greater than 99.0% probability and contained at least two identified peptides. Protein probabilities were assigned by the Protein Prophet algorithm (*Nesvizhskii et al., 2003*). Proteins that contained similar peptides and could not be differentiated based on MS/MS analysis alone were grouped to satisfy the principles of parsimony. Proteins sharing significant peptide evidence were grouped into clusters. The resulting normalized spectral counts from the input NE, each RAC substrate (WT, upDEL, dnDEL, 2xDEL, and APT Only control) were obtained, and those values were used to calculate background-corrected levels of enrichment relative to NE (($NSC_{RAC} - NSC_{APT}$)/$NSC_{NE}$). We used those calculations to construct the heatmap shown in *Figure 4B* and generate the counts shown in *Figure 4C*, by including proteins with experimental LC-MS/MS evidence of being members of early spliceosomes (E-complex or 17S U2 snRNP *Makarov et al., 2012*; *Sharma et al., 2008*; *Cvitkovic and Jurica, 2013*), and that had a positive value of background-corrected levels of enrichment relative to NE in at least 1 of the 4 RAC substrates tested. See *Supplementary file 4*.

### Data-independent acquisition NanoLC MS/MS analysis

Peptide mixtures were analyzed by nanoflow liquid chromatography-tandem mass spectrometry (nanoLC-MS/MS) using a nano-LC chromatography system (UltiMate 3000 RSLCnano, Dionex), coupled on-line to a Thermo Orbitrap Eclipse mass spectrometer (Thermo Fisher Scientific, San Jose, CA) through a nanospray ion source. A direct injection method is used onto an analytical column; Aurora (75 µm×25 cm, 1.6 µm) from (ionopticks). After equilibrating the column in 98% solvent A (0.1% (v/v) formic acid in water) and 2% solvent B (0.1% (v/v) formic acid in acetonitrile (ACN)), the samples (2 µL in solvent A) were injected (300 nL/min) by gradient elution onto the C18 column as follows: isocratic at 2% B, 0–10 min; 2 to 27% 10–98 min, 27 to 45% B, 98–102 min; 45 to 90% B, 102–103 min; isocratic at 90% B, 103–104 min; 90 to 15%, 104–106 min; 15 to 90% 106–108 min; isocratic for 2 min; 90 to 2%, 110–112 min; and isocratic at 2% B, till 120 min.

All LC-MS/MS data were acquired using an Orbitrap Eclipse in positive ion mode using a data-independent acquisition (DIA) method with a 16 Da window from 400 to 1000 and a loop time of 3 s. The survey scans (m/z 350–1500) were acquired in the Orbitrap at 60,000 resolution (at m/z=400) in centroid mode, with a maximum injection time of 118 ms and an AGC target of 100,000 ions. The S-lens RF level was set to 60. Isolation was performed in the quadrupole, and HCD MS/MS acquisition was performed in profile mode using the Orbitrap at a resolution of 30,000 using the following settings: collision energy = 33%, IT 54 ms, AGC target = 50,000. These conditions were duplicated to create six gas-phase fractions of the NE sample using 4 Da fully staggered windows in 100 m/z increments from 400 to 1000 m/z, as described (*Searle et al., 2020*).

### DIA database searching

The raw data was demultiplexed to mzML with 10 ppm accuracy after peak picking in MSConvert (*Chambers et al., 2012*). The resulting mzML files were searched in MSFragger (*Kong et al., 2017*) and quantified via DIA-NN (https://github.com/vdemichev/DiaNN; *Demichev, 2025*) using the following settings: peptide length range 7–50, protease set to Trypsin, two missed cleavages, three3 variable modifications, clip N-term M on, fixed C carbamidomethylation, variable modifications of methionine oxidation and n-terminal acetylation, MS1 and MS2 accuracy set to 20 ppm, 1% FDR, and DIANN quantification strategy set to Robust LC (high accuracy). The files were searched against a database of human proteins acquired from Uniprot (18 December, 2023). The gas-phase fractions were used only to generate the spectral library, which was used for analysis of the individual samples.

Statistical analysis was performed using Fragpipe-Analyst (*Hsiao et al., 2024*) using an R script based on the ProteinGroup.txt file produced by DIA-NN. First, contaminant proteins, reverse sequences, and proteins identified 'only by site' were filtered out. In addition, proteins identified by only a single peptide and those that are not consistently identified or quantified across the same condition are also removed. The DIA data were converted to a $\log_2$ scale, samples were grouped by conditions, and missing values were not imputed. Protein-wise linear models combined with empirical Bayes statistics were used for the differential expression analyses. The limma package from R Bioconductor was used to generate a list of differentially expressed proteins for each pairwise comparison. A cutoff of the adjusted *p*-value of 0.05 (Benjamini-Hochberg method) and an absolute $\log_2$ fold change of 1 have been applied to determine significantly regulated proteins in each pairwise comparison. We also used the E/U2 list to focus our analysis on early spliceosome proteins and expanded this to include an RBP list (*Sasse et al., 2025*) to assess non-spliceosome-annotated RBP interactions. The E/U2 or RBPs shown in *Figure 4E* were included if (1) spectral counts were detected in both the numerator and denominator RAC substrates, (2) log2 fold change $\geq |0.2|$ and *p*-value was <0.01 in at least one of the three comparison groups, and (3) data were present for each comparison (no zero or undetectable values). See *Supplementary file 5*.

### Materials availability statement

Newly created materials, such as plasmids, are available upon request to the corresponding author.

## Acknowledgements

We thank Manuel Ares Jr. (UCSC) for helpful discussion regarding the experiments in *S. cerevisiae* and for piloting an early version of the experiment. We thank Louise Prakash (UTMB) for allowing us

to use equipment that enabled us to perform the experiments in *S. cerevisiae*, and Robert E Johnson of the Prakash lab for technical assistance. We thank Steve Widen (UTMB, retired) for technical assistance with the QKI and SF1 knockdown RNA-seq datasets. We thank Mariano Garcia-Blanco and Chloe Nagasawa from Mariano's lab for discussion and technical suggestions on RNA affinity chromatography. We also thank Eric Van Nostrand (Baylor College of Medicine) and Manuel Ares Jr. for pre-submission review of the manuscript. Molecular graphics and analyses performed with UCSF ChimeraX, developed by the Resource for Biocomputing, Visualization, and Informatics at the University of California, San Francisco, with support from National Institutes of Health R01-GM129325 and the Office of Cyber Infrastructure and Computational Biology, National Institute of Allergy and Infectious Diseases.

## Additional information

### Funding

| Funder | Grant reference number | Author |
|---|---|---|
| National Institute of General Medical Sciences | R35GM151324 | William S Fagg |
| Cancer Prevention and Research Institute of Texas | RP190682 | William K Russell |

The funders had no role in study design, data collection and interpretation, or the decision to submit the work for publication.

### Author contributions

Karen Larisssa Pereira de Castro, William K Russell, Formal analysis, Investigation, Methodology; Jose M Abril, Investigation, Methodology; Kuo-Chieh Liao, Resources; Haiping Hao, John Paul Donohue, Investigation; William S Fagg, Conceptualization, Resources, Data curation, Formal analysis, Supervision, Funding acquisition, Validation, Investigation, Visualization, Methodology, Writing – original draft, Project administration, Writing – review and editing

### Author ORCIDs

Haiping Hao ⓘ https://orcid.org/0000-0002-8826-1568
William S Fagg ⓘ https://orcid.org/0000-0003-2909-1666

Reviewer #2 (Public review): https://doi.org/10.7554/eLife.103167.3.sa1
Reviewer #3 (Public review): https://doi.org/10.7554/eLife.103167.3.sa2
Author response https://doi.org/10.7554/eLife.103167.3.sa3

## Additional files

### Supplementary files

Supplementary file 1. Vast-tools splicing analysis datafile from ENCODE RNA-seq data.

Supplementary file 2. rMATS splicing analysis datafile from ENCODE RNA-seq data comparing shQKI to shControl.

Supplementary file 3. rMATS splicing analysis datafile from ENCODE RNA-seq data comparing shSF1 to shControl.

Supplementary file 4. LC-MS/MS datafile from RNA affinity chromatography in the presence of ATP.

Supplementary file 5. LC-MS/MS datafile from RNA affinity chromatography in the absence of ATP.

Supplementary file 6. Transcript abundance analysis of RNA-seq data from BY4741 with QKI5 expression compared to parental control.

Supplementary file 7. Splicing analysis of RNA-seq data from BY4741 with QKI5 expression compared to parental control.

Supplementary file 8. Oligonucleotide sequence info file.

MDAR checklist

## Data availability
New RNA-seq data files have been uploaded to GEO under accession GSE273838. All data generated or analyzed in this study are also included in the manuscript and supporting files; source data files have been provided for all figures.

The following dataset was generated:

| Author(s) | Year | Dataset title | Dataset URL | Database and Identifier |
|---|---|---|---|---|
| de Castro KL, Johnson RE, Abril J, Kuo-Chieh L, Widen SG, Hao H, Donohue JP, Russell WK, Fagg WS | 2024 | An ancient competition for the conserved branchpoint sequence influences physiological and evolutionary outcomes in splicing | https://www.ncbi.nlm.nih.gov/geo/query/acc.cgi?acc=GSE273838 | NCBI Gene Expression Omnibus, GSE273838 |

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
