## [Editor Report · eLife Assessment]

This **important** manuscript provides insights into the competition between Splicing Factor 1 (SF1) and Quaking (QKI) for binding at the ACUAA branch point sequence in a model intron, regulating exon inclusion. The study employs **convincing**, rigorous transcriptomic, proteomic, and reporter assays, with both mammalian cell culture and yeast models.

---

## [Referee Report · Reviewer #2 (Public review)]

Summary:

In this manuscript, Pereira de Castro and coworkers are studying potential competition between a more standard splicing factor SF1 and an alternative splicing factor called QK1. This is interesting because they bind to overlapping sequence motifs and could potentially have opposing effects on promoting the splicing reaction. To test this idea, the authors KD either SF1 or QK1 in mammalian cells and uncover several exons whose splicing regulation follows the predicted pattern of being promoted for splicing by SF1 and repressed by QK1. Importantly, these have introns enriched in SF1 and QK1 motifs. The authors then focus on one exon in particular with two tandem motifs to study the mechanism of this in greater detail and their results confirm the competition model. Mass spec analysis largely agrees with their proposal; however, it is complicated by apparently quick transition of SF1 bound complexes to later splicing intermediates. An inspired experiment in yeast shows how QK1 competition could potentially have a determinental impact on splicing in an orthogonal system. Overall these results show how splicing regulation can be achieved by competition between a "core" and alternative splicing factor and provide additional insight into the complex process of branch site recognition. The manuscript is exceptionally clear and the figures and data very logically presented. The work will be valuable to those in the splicing field who are interested in both mechanism and bioinformatics approaches to deconvolve any apparent "splicing code" being used by cells to regulate gene expression.

Strengths:

(1) The main discovery of the manuscript involving evidence for SF1/QK1 competition is quite interesting and important for this field. This evidence has been missing and may change how people think about branch site recognition.

(2) The experiments and the rationale behind them are clearly and logically presented.

(3) The experiments are carried out to a high standard and well-designed controls are included.

(4) The extrapolation of the result to yeast in order to show the potentially devastating consequences of QK1 competition was creative and informative.

Weaknesses:

Overall the weaknesses are relatively minor and involve cases where conclusions could potentially have been strengthened with additional experimentation. For example, pull-down of the U2 snRNP could be strengthened by detection of the snRNA whereas the proteins may themselves interact with these factors in the absence of the snRNA. In addition the discussion is a bit speculative given the data, but compelling nonetheless.

---

## [Referee Report · Reviewer #3 (Public review)]

Summary:

In this manuscript the authors were trying to establish whether competition between the RNA binding proteins SF1 and QKI controlled splicing outcomes. These two proteins have similar binding sites and protein sequences, but SF1 lacks a dimerization motif and seems to bind a single version of the binding sequence. Importantly, these binding sequences correspond to branchpoint consensus sequences, with SF1 binding leading to productive splicing, but QKI binding leading instead to association with paraspeckle proteins. They show that in human cells SF1 generally activates exons and QKI represses, and a large group of the jointly regulated exons (43% of joint targets) are reciprocally controlled by SF1 and QKI. They focus on one of these exons RAI14 that shows this reciprocal pattern of regulation, and has 2 repeats of the binding site that make it a candidate for joint regulation, and confirm regulation within a minigene context. The authors used assembly of proteins within nuclear extracts to explain the effect of QKI versus SF1 binding. Finally the authors show that expression of QKI is lethal in yeast, and causes splicing defects.

How this fits in the field. This study is interesting and provides a conceptual advance by providing a general rule how SF1 and QKI interact with relation to binding sites, and the relative molecular fates followed, so is very useful. Most of the analysis seems to focus on one example, but the choice of this example was carefully explained in the text. The molecular analysis and global work significantly adds to the picture from the previously published paper about NUMB joint regulation by QKI and SF (Zong et al, cited in text as reference 50, that looked at SF1 and QKI binding in relation to a duplicated binding site/branchpoint sequence in NUMB).

Strengths:

The data presented are strong and clear. The ideas discussed in this paper are of wide interest, and present a simple model where two binding sites generates a potentially repressive QKI response, whereas exons that have a single upstream sequence are just regulated by SF1. The assembly of splicing complexes on RNAs derived from RAI14 in nuclear extracts, followed by mass spec gave interesting mechanistic insight into what was occurring as a result of QKI versus SF1 binding.

Weaknesses:

The authors have addressed the previous weaknesses of the study, resulting in a much stronger manuscript

---

## [Author Response]

The following is the authors’ response to the original reviews.

**eLife Assessment**
This important manuscript provides insights into the competition between Splicing Factor 1 (SF1) and Quaking (QKI) for binding at the ACUAA branch point sequence in a model intron, regulating exon inclusion. The study employs rigorous transcriptomic, proteomic, and reporter assays, with both mammalian cell culture and yeast models. Nevertheless, while the data are convincing, broadening the analysis to additional exons and narrowing the manuscript's title to better align with the experimental scope would strengthen the work.
**Public Reviews:**

**Reviewer #1 (Public review):**
In this manuscript, the authors aimed to show that SF1 and QKI compete for the intron branch point sequence ACUAA and provide evidence that QKI represses inclusion when bound to it.Major strengths of this manuscript include:(1) Identification of the ACUAA-like motif in exons regulated by QKI and SF1.(2) The use of the splicing reporter and mutant analysis to show that upstream and downstream ACUAAC elements in intron 10 of RAI are required for repressing splicing.(3) The use of proteomic to identify proteins in C2C12 nuclear extract that binds to the wild type and mutant sequence.(4) The yeast studies showing that ectopic lethality when Qki5 expression was induced, due to increased mis-splicing of transcripts that contain the ACUAA element.The authors conclusively show that the ACUAA sequence is bound by QKI and provide strong evidence that this leads to differences in exons inclusion and exclusion. In animal cells, and especially in human, branchpoint sequences are degenerate but seem to be recognized by specific splicing factors. Although a subset of splicing factors shows tissue-specific expression patterns most don't, suggesting that yet-to-be-identified mechanisms regulate splicing. This work suggests that an alternate mechanism could be related to the binding affinity of specific RNA binding factors for branchpoint sequences coupled with the level of these different splicing factors in a given cell.

We thank the reviewer for the positive comments.

**Reviewer #2 (Public review):**
Summary:In this manuscript, Pereira de Castro and coworkers are studying potential competition between a more standard splicing factor SF1, and an alternative splicing factor called QK1. This is interesting because they bind to overlapping sequence motifs and could potentially have opposing effects on promoting the splicing reaction. To test this idea, the authors KD either SF1 or QK1 in mammalian cells and uncover several exons whose splicing regulation follows the predicted pattern of being promoted for splicing by SF1 and repressed by QK1. Importantly, these have introns enriched in SF1 and QK1 motifs. The authors then focus on one exon in particular with two tandem motifs to study the mechanism of this in greater detail and their results confirm the competition model. Mass spec analysis largely agrees with their proposal; however, it is complicated by the apparently quick transition of SF1-bound complexes to later splicing intermediates. An inspired experiment in yeast shows how QK1 competition could potentially have a detrimental impact on splicing in an orthogonal system. Overall, these results show how splicing regulation can be achieved by competition between a "core" and alternative splicing factor and provide additional insight into the complex process of branch site recognition. The manuscript is exceptionally clear and the figures and data are very logically presented. The work will be valuable to those in the splicing field who are interested in both mechanism and bioinformatics approaches to deconvolve any apparent "splicing code" being used by cells to regulate gene expression. Criticisms are minor and the most important of them stem from overemphasis on parts of the manuscript on the evolutionary angle when evolution itself wasn't analyzed per se.

We thank the reviewer for the positive comments and very clear and fair critical points.

Strengths:(1) The main discovery of the manuscript involving evidence for SF1/QK1 competition is quite interesting and important for this field. This evidence has been missing and may change how people think about branch site recognition.(2) The experiments and the rationale behind them are exceptionally clearly and logically presented. This was wonderful!

Thank you so much. We felt the overall flow of the paper and data make for a nice “story” that conveys a relatively easy-to-understand explanation for a complex subject.

(3) The experiments are carried out to a high standard and well-designed controls are included.(4) The extrapolation of the result to yeast in order to show the potentially devastating consequences of the QK1 competition was very exciting and creative.

We agree this is a very exciting result and finding! Thanks.

Weaknesses:Overall the weaknesses are relatively minor and involve cases where clarification is necessary, some additional analysis could bolster the arguments, and suggestions for focusing the manuscript on its strengths.(1) The title (Ancient...evolutionary outcomes), abstract, and some parts of the discussion focus heavily on the evolutionary implications of this work. However, evolutionary analysis was not performed in these studies (e.g., when did QK1 and SF1 proteins arise and/or diverge? How does this line up with branch site motifs and evolution of U2? Any insight from recent work from Scott Roy et al?). I think this aspect either needs to be bolstered with experimental work/data or this should be tamped down in the manuscript. I suggest highlighting the idea expressed in the sentence "A nuanced implication of this model is that loss-of-function...". To me, this is better supported by the data and potentially by some analysis of mutations associated with human disease.

We have revised the title and dampened the evolutionary aspects of the previous version of the manuscript.

(2) One paper that I didn't see cited was that by Tanackovic and Kramer (Mol Biol Cell 2005). This paper is relevant because they KD SF1 and found it nonessential for splicing in vivo. Do their results have implications for those here? How do the results of the KD compare? Could QK1 competition have influenced their findings (or does their work influence the "nuanced implication" model referenced above?)?

This is an interesting point, and thank you for the suggestion. We have now included a brief description of this study in the Introduction of the revised manuscript and do note that the authors measured intron retention of a beta globin reporter and SF3A1, SF3A2, and SF3A3 during SF1 knockdown, but did not detect elevated unspliced RNA in these targets.

(3) Can the authors please provide a citation for the statement "degeneracy is observed to a higher degree in organisms with more alternative splicing"? Does recent evolutionary analysis support this?

We have removed the statement, as it did not add much to the content and I am not sure I can state the concept I was attempting to convey in a simple manner with few citations.

(4) For the data in Figure 3, I was left wondering if NMD was confounding this analysis. Can the authors respond to this and address this concern directly?

We have not measured if the reporters used in Figure 3 produce protein(s). Presumably, though, all spliced reporter RNA would be degraded equally (the included/skipped isoforms’ “reading frames” are not altered from one another). This would not be case for unspliced nuclear reporter RNA, however. Given this difference, and that our analysis can not resolve the subcellular localization of the different reporter species, we have removed the measurement of and subsequent results describing unspliced reporter RNA from Figure 3.

(5) To me, the idea that an engaged U2 snRNP was pulled down in Figure 4F would be stronger if the snRNA was detected. Was that able to be observed by northern or primer extension? Would SF1 be enriched if the U2 snRNA was degraded by RNaseH in the NE?

We did not measure any co-associating RNAs in this experimental approach, but agree that this approach would strengthen the evidence for it.

(6) I'm wondering how additive the effects of QK1 and SF1 are... In Figure 2, if QK1 and SF1 are both knocked down, is the splicing of exon 11 restored to "wt" levels?

This is an interesting question that we were unfortunately unable to address experimentally here.

(7) The first discussion section has two paragraphs that begin "How does competition between SF1..." and "Relatively little is known about how...". I found the discussion and speculation about localization, paraspekles, and lncRNAs interesting but a bit detracting from the strengths of the manuscript. I would suggest shortening these two paragraphs into a single one.

We have revised the Discussion.

**Reviewer #3 (Public review):**
Summary:In this manuscript, the authors were trying to establish whether competition between the RNA-binding proteins SF1 and QKI controlled splicing outcomes. These two proteins have similar binding sites and protein sequences, but SF1 lacks a dimerization motif and seems to bind a single version of the binding sequence. Importantly, these binding sequences correspond to branchpoint consensus sequences, with SF1 binding leading to productive splicing, but QKI binding leading instead to association with paraspeckle proteins. They show that in human cells SF1 generally activates exons and QKI represses, and a large group of the jointly regulated exons (43% of joint targets) are reciprocally controlled by SF1 and QKI. They focus on one of these exons RAI14 that shows this reciprocal pattern of regulation, and has 2 repeats of the binding site that make it a candidate for joint regulation, and confirm regulation within a minigene context. The authors used the assembly of proteins within nuclear extracts to explain the effect of QKI versus SF1 binding. Finally, the authors show that the expression of QKI is lethal in yeast, and causes splicing defects.How this fits in the field. This study is interesting and provides a conceptual advance by providing a general rule on how SF1 and QKI interact in relation to binding sites, and the relative molecular fates followed, so is very useful. Most of the analysis seems to focus on one example, although the molecular analysis and global work significantly add to the picture from the previously published paper about NUMB joint regulation by QKI and SF (Zong et al, cited in text as reference 50, that looked at SF1 and QKI binding in relation to a duplicated binding site/branchpoint sequence in NUMB).

Thank you for the encouraging remarks.

Strengths:The data presented are strong and clear. The ideas discussed in this paper are of wide interest, and present a simple model where two binding sites generate a potentially repressive QKI response, whereas exons that have a single upstream sequence are just regulated by SF1. The assembly of splicing complexes on RNAs derived from RAI14 in nuclear extracts, followed by mass spec gave interesting mechanistic insight into what was occurring as a result of QKI versus SF1 binding.Weaknesses:I did not think the title best summarises the take-home message and could be perhaps a bit more modest. Although the authors investigated splicing patterns in yeast and human cells, yeast do not have QKI so there is no ancient competition in that case, and the study did not really investigate physiological or evolutionary outcomes in splicing, although it provides interesting speculation on them. Also as I understood it, the important issue was less conserved branchpoints in higher eukaryotes enabling alternative splicing, rather than competition for the conserved branchpoint sequence. So despite the the data being strong and properly analysed and discussed in the paper, could the authors think whether they fit best with the take-home message provided in the title? Just as a suggestion (I am sure the authors can do a better job), maybe "molecular competition between variant branchpoint sequences predict physiological and evolutionary outcomes in splicing"?

Thank you for this point (Reviewer 2 had a similar comment) and the suggestion. We have revised the title.

Although the authors do provide some global data, most of the detailed analysis is of RAI14. It would have been useful to examine members of the other quadrants in Figure 1C as well for potential binding sites to give a reason why these are not co-regulated in the same way as RAI14. How many of the RAI14 quadrants had single/double sites (the motif analysis seemed to pull out just one), and could one of the non-reciprocally regulated exons be moved into a different quadrant by addition or subtraction of a binding site or changing the branchpoint (using a minigene approach for example).

This is an interesting point that we have considered. Our intent with the focus on RAI14 was to use a naturally occurring intron bps with evidence of strong QKI binding that did not require a high degree of sequence manipulation or engineering.

**Recommendations for the authors:**

**Reviewer #1 (Recommendations for the authors):**
(1) Most of my recommendations are really centered on the figures. In their current state, they detract from the data shown and could be improved: I recommend the authors use a uniform font. For example, Figure 1E and F have at least three different fonts of varying sizes making it very messy. In Figure 1C, the authors could bold the Ral14 ex11 or simply indicate that the blue is this exon in the legend, thus removing the text from this very busy graph. In Figure 4F, I would recommend, having all the labels the same size and putting those genes of interest like Sf3a1 in bold. This could also be done in Figure 4E.

Thank you for the suggestion and we have edited these (FYI the font in Fig’s 1E and 1F were from the rMAPS default output, but I agree, it gives a sloppy appearance).

(2) In Figures 4D and 4G, is there QKI binding to the downstream deletion mutant after 30 minutes? Also, in Figure 4G, are these all from the same blot? The band sizes seem to be very different between lanes. If these were not on the same blot, the original gels should be submitted.

A small amount of Qki appears to be binding after 30 min. All lanes/blots are from the same gels/membranes; see new Supplemental Figure 4 for the original (uncropped) images of the blots.

(3) The authors should indicate, the source and concentration of the antibodies used for their WB. They should also indicate the primers used for RT-PCRs.

We have revised the methods to include the antibody information and have uploaded a supplemental table 8 with all oligonucleotide sequences used (which I (Sam Fagg) neglected to do initially, so that’s my bad).

**Reviewer #2 (Recommendations for the authors):**
(1) This may come down to the author's preference but branch point and branch site are frequently two words, not a single compound word (branch point vs. branchpoint). In addition, the authors may want to use branchsite with the abbreviation BS more frequently since they often don't describe the specific point of branching, and bp and bps could be confused for the more frequent abbreviations for base pair(s).

Good suggestion; we have edited the text accordingly.

(2) In general the addition of page numbers and line numbers to the manuscript would greatly aid reviewers!

Point taken…

(3) Introduction; "...under normal growth conditions they are efficiently spliced". I would say MOST introns in yeast are efficiently spliced. This is definitely not universal.

Text edited to indicate that most are efficiently spliced.

(4) Introduction; " recognition of the bps by SF1 (mammals) (20)". The choice of reference 20 is an odd one here. I think the Robin Reed and Michael Rosbash paper was the first to show SF1 was the human homolog of BBP.

Got it, thanks (added #14 here and kept #20 also since it shows the structure of SF1 in complex with a UACUAAC bps.)

(5) Results; "QK1 and SF1 co-regulate.."; it may be useful for the reader if you could explain in more detail why exon inclusion and intron retention are expected outcomes for QK1 knockdown and vice versa for SF1. The exon inclusion here is more obvious than the intron retention phenotype. (In other words, if more exons are included shouldn't it follow that more introns are removed?)

We explain the expected results for exon inclusion in the Introduction and this paragraph of the Results. Although we have observed more intron retention under QKI loss-of-function approaches before, I am uncertain where the reviewer sees that we indicate any expected result for intron retention from either QKI or SF1 knockdown. I believe the statement you refer to might be on line 162 and starts with: “Consistent with potentially opposing functions in splicing…” ?

Also, I agree that if SF1 is a “splicing activator,” one might expect more IR in its absence (but this is not the case; there is, in fact, less), but nonetheless, the opposite outcome is observed with QKI knockdown (more IR). It is unclear why this is the case, and we did not investigate it.

(6) Results; "QK1 and SF1 co-regulate.."; "Thus the most highly represented set.." To me, the most highly represented set is those which are not both QK1-repressed and SF1-activated. Does this indicate that other factors are involved at most sites than simple competition between these two?

We have revised the sentence in question to include the text “by quadrant” in order to convey our meaning more precisely.

(7) Throughout the manuscript, 5 apostrophes and 3 apostrophes are used instead of 5 prime symbols and 3 prime symbols.

Thank you for pointing that out. We have fixed each instance of this.

(8) Sometimes SF1 is written as Sf1. (also Tatsf1)

This was a mouse/human gene/protein nomenclature error that we have fixed; thank you for pointing this out.

(9) You may want to make sure that figures are labeled consistently with the manuscript text. In Figure 1B, it is RI rather than IR. In Figure 4 it is myoblast NE rather than C2C12 nuclear extract.

We have fixed these, checked for other examples, and where relevant, edited those too.

(10) I think Figure 1A could be improved by also including a depiction of the domain arrangements of SF1 and QK1.

Done.

(11) I was a bit confused with all the lines in Figure 1E and 1F. What is the difference between the log (pVal) and upregulated plots? Can these figures be simplified or explained more thoroughly?

Based on this comment and one from Reviewer 1, we have slightly revised the wording (and font) on the output, which hopefully clarifies. These are motif enrichment plots generated by rMAPS (Refs 61 and 62) analysis of rMATS (Ref 60) data for exons more included (depicted by the red lines) or more skipped (depicted by the blue lines) compared to control versus a “background” set of exons that are detectable but unchanged. The -log_10_ is P-value (dotted line) indicates the significance of exons more included in shRNA treatment vs control shRNA (previously read “upregulated”) compared to background exons that are detectable but unchanged; the solid lines indicate the motif score; these are described in the references indicated.

(12) Figure 1B, it is a bit hard to conclude that there is more AltEx or "RI/IR" in one sample vs. the other from these plots since the points overlay one another. Can you include numbers here?

Added (and deleted Suppl Fig S1, which was simply a chart showing the numbers).

(13) How was PSI calculated in Figure 2A?

VAST-tools (we state this in the legend in the revised version).

You may want to include rel protein (or the lower limit of detection) for Figure 2B to be consistent with 2C. Why is KD of SF1 so poor and variable between 2C and 2D?

We have not investigated this, but these blots show an optimized result that we were able to obtain for the knockdown in each cell type. It may be that HEK293 cells (Fig 2B) have a stronger requirement for SF1 than C2C12 cells…? I would argue that it is not necessarily “poor” in Fig 2C, as we observe ~70% depletion of the protein.

Why are two bands present in the gel?

Two to three isoforms of SF1 are present in most cell types.

A good (or bad, really) example of an SF1 western blot and knockdown of ~35% in K562 or ~45% in HepG2 can also be seen on the ENCODE project website, for reference:

https://www.encodeproject.org/documents/6001a414-b096-4073-94ff-3af165617eb5/@@download/attachment/SF1_BGKLV28-49.pdf

By comparison, I think ours are much more cosmetically pleasing, and our knockdown (especially in C2C12) is much more efficient.

(14) Figure 3, The asterisk refers to a cryptic product. Can the uaAcuuuCAG be used as a branch point? Presumably the natural 3' SS is now too close so this would result in activation of a downstream 3'SS?

We did not pursue determining the identity of this minor and likely artefactual product, but we (and others) have observed a similar phenomenon when using splicing reporter-based mutational approaches.

(15) For the methods. The "RNA extraction, RT -PCR,..." subheading needs to be on its own line. Please add (w/v) or (v/v) to percentages where appropriate. Please convert ug to the symbol for "micro".

Thank you, we have made these changes.

(16) In Figure 4B, the text here and legend are microscopic. Even with reading glasses, I couldn't make anything out!

We have increased the font sizes for the text and scale bar…when referring to “legend” does the reviewer mean the scale bar?

(17) As a potential discussion item, it is worth noting that SF1 could also repress splicing if it could either not engage with U2AF or be properly displaced by U2 snRNP so the snRNA could pair. I was wondering if QK1 could similarly be activating if it could engage with U2AF. I'm unsure if this could be tested by domain swaps (and is beyond the scope of this paper). It just may be worth speculating about.

Good point and suggestion…we are looking into this.

**Reviewer #3 (Recommendations for the authors):**
(1) Is the reference in the text to Figure 5F correct for actin splicing (this is just before the discussion)?

I see references several lines up from this, but I do not see a reference just before the discussion…?

(2) I was not sure why the minigene experiments showed such high levels of intron retention that seemed to be impacted also by deletion of the branchpoint sequences, and suggest that the two branchpoints are not equal in strength.

Neither were we, but Reviewer 2 has suggested that degradation of the spliced products could be rapid (NMD substrates) which could complicate the interpretation of what appears to be higher levels of intron retention. Given the possibility that this could be a non-physiological artefact, we have removed the measurement of unspliced reporter and now only show the spliced products (equally subject to degradation) and report their percent inclusion.